# Unraveling the Mechanism of Action, Binding Sites, and Therapeutic Advances of CFTR Modulators: A Narrative Review

**DOI:** 10.3390/cimb47020119

**Published:** 2025-02-11

**Authors:** Debora Baroni

**Affiliations:** Istituto di Biofisica, Consiglio Nazionale delle Ricerche (CNR), Via De Marini, 6, 16149 Genova, Italy; debora.baroni@ibf.cnr.it

**Keywords:** cystic fibrosis, CFTR, CFTR small molecule modulators, potentiators, correctors

## Abstract

Cystic fibrosis (CF) is a recessive genetic disease caused by mutations in the cystic fibrosis transmembrane conductance regulator (CFTR) protein, a chloride and bicarbonate channel localized on the plasma membrane of epithelial cells. Over the last three decades, high-throughput screening assays have been extensively employed in identifying drugs that target specific defects arising from CFTR mutations. The two main categories of such compounds are potentiators, which enhance CFTR gating by increasing the channel’s open probability, and correctors, which improve CFTR protein folding and trafficking to the plasma membrane. In addition to these, other investigational molecules include amplifiers and stabilizers, which enhance the levels and the stability of CFTR on the cell surface, and read-through agents that promote the insertion of correct amino acids at premature termination codons. Currently, four CFTR modulators are clinically approved: the potentiator ivacaftor (VX-770), either as monotherapy or in combination with the correctors lumacaftor (VX-809), tezacaftor (VX-661), and elexacaftor (VX-445). Among these, the triple combination VX-445/VX-661/VX-770 (marketed as Trikafta^®^ in the US and Kaftrio^®^ in Europe) has emerged as the most effective CFTR modulator therapy to date, demonstrating significant clinical benefits in phase III trials for patients with at least one F508del CFTR allele. Despite these advancements, the mechanisms of action and binding sites of these modulators on CFTR have only recently begun to be elucidated. A deeper understanding of these mechanisms could provide essential insights for developing more potent and effective modulators, particularly in combination therapies. This narrative review delves into the mechanism of action, binding sites, and combinatorial effects of approved and investigational CFTR modulators, highlighting ongoing efforts to broaden therapeutic options for individuals with CF.

## 1. Introduction

Cystic fibrosis (CF) is a life-threatening autosomal recessive disorder affecting approximately 50,000 people in Europe and over 85,000 worldwide. Once primarily considered a pediatric condition, advancements in treatments have significantly improved the prognosis, with the predicted mean survival of newborns with CF in 2019 at 50 years [1]. Morbidity and mortality in people with CF are predominantly caused by progressive obstructive lung disease, characterized by cycles of airway infection and inflammation, resulting in bronchiectasis, tissue remodeling, and a gradual decline in lung function. Besides the respiratory system, CF affects various other organs, including the exocrine pancreas, intestine, vas deferens, and sweat glands, necessitating comprehensive therapeutic approaches [2,3]. The symptoms of CF arise from an imbalance in ion and water regulation within the epithelial tissues of secretory organs. This imbalance results from the loss of function of the cystic fibrosis transmembrane conductance regulator (CFTR) protein, which acts as a cAMP-regulated chloride (Cl^−^) and bicarbonate (HCO_3_^−^) anion channel localized on the plasma membrane of epithelial cells [4,5]. Mutations in the CFTR gene lead to the absence or dysfunction of the CFTR protein, ultimately resulting in the production of thick, sticky mucus that can cause severe respiratory and digestive issues. In the airways, this dense secretion is not readily removed due to a decrease in the height of the periciliary liquid layer, a thin watery layer lining the apical surface of epithelial cells which is essential for mucociliary clearance [6]. Additionally, changes in pH caused by reduced bicarbonate secretion impair the proper unfolding of mucins, leading to more compact mucus that is harder to expel [7,8]. Coupled with altered innate immune responses, such as the loss of antimicrobial peptide function and the overactive but ineffective response of infiltrating neutrophils, this creates an environment that is highly conducive to opportunistic microorganisms. Chronic colonization with pathogens such as *Pseudomonas aeruginosa*, *Burkholderia cepacia*, and *Staphylococcus aureus* increases as patients age, with over 80% of individuals colonized by at least one pathogen by age 20 [9]. Chronic infection correlates with more rapid disease progression and a worse prognosis, emphasizing the importance of early treatment to prevent persistent infections.

CFTR mutations are categorized into six classes based on their impact on the protein biogenesis and function [10,11]. Class I mutations result in no CFTR protein production (e.g., G542X, W1282X), and class II mutations, such as F508del, the most frequent among CFTR mutations, cause defective protein folding and trafficking. Class III mutations impair channel gating (e.g., G551D), and class IV mutations result in defective conductance (e.g., R117H). Class V mutations lead to reduced CFTR protein synthesis, often due to splicing defects (e.g., c.3718-2477C>T). Class VI mutations affect CFTR stability, often due to missense or truncation mutations near the C-terminus (e.g., S492F or Q1412X). It is noteworthy that many mutations are pleiotropic and could therefore be assigned to more than one class. For example, F508del is primarily characterized by folding and processing defects (class II) that prevent it from reaching the PM, but it also exhibits gating defects (class III) and reduced stability once at the PM (class VI).

The advent of CFTR modulators has revolutionized the treatment of CF by directly targeting the underlying molecular defects in the CFTR protein [12,13,14]. These small-molecule drugs include potentiators, which enhance the gating function of CFTR by increasing the channel-open probability (PO), and correctors, which improve the folding and trafficking of mutant CFTR to the PM. In addition, emerging investigational therapeutics, including amplifiers, stabilizers, read-through agents, and gene- and RNA-based treatments, have shown significant potential in addressing the basic defects of CFTR. Figure 1 provides a schematic overview of the mechanism of action of currently available and investigational drugs aimed at targeting the basic defects of CFTR. Despite this progress, significant challenges remain in the treatment of CF, particularly for patients with mutations that do not respond to currently approved CFTR modulators, leaving them without targeted therapeutic options. This review aims to explore the structure and function of the CFTR protein, the impacts of the different CFTR mutation classes, and the mechanisms of action and binding sites of both approved and investigational CFTR modulators. The goal is to pave the way for the rational design of next-generation molecules that could broaden the therapeutic landscape and increase the number of CFTR mutations responsive to available treatments.

## 2. Literature Search Strategy

An extensive literature search was conducted across the PubMed database to achieve the objectives of this review. The search employed a combination of key terms such as “CFTR modulators”, “CFTR binding sites”, “potentiators”, “correctors”, and “therapeutic strategies in cystic fibrosis”, and covered the period from 1989 to the present to ensure the inclusion of the most up-to-date and relevant information. Only studies published in English were considered, and the focus was placed on peer-reviewed articles to guarantee scientific rigor. The selected studies were required to provide experimental data, clinical trial results, or substantial review insights relevant to elucidating the mechanisms of action and binding sites of CFTR modulators.

The studies that met the inclusion criteria were carefully analyzed and organized into key thematic areas, including the mechanisms of action and binding sites of CFTR modulators, as well as the therapeutic potential of both approved and investigational CFTR modulators. Important findings, methodologies, and conclusions from each study were extracted to ensure that only the most pertinent and scientifically robust studies were incorporated into the review. This approach allowed for a balanced presentation of both approved therapies and investigational drugs, offering a comprehensive overview of the current understanding of CFTR modulator research.

## 3. CFTR Structure

The cystic fibrosis transmembrane conductance regulator (CFTR) protein belongs to the ATP-binding cassette (ABC) transporter family; however, unlike other members, it functions as an ion channel rather than a transporter. CFTR consists of 1480 amino acids and is organized into five main domains: two membrane-spanning domains (MSDs), two nucleotide-binding domains (NBDs), and a unique regulatory domain (RD) that distinguishes CFTR from the other ABC transporters. The MSDs provide the pathway for chloride ion transport across the cell membrane, while the NBDs regulate the channel’s gating through ATP binding and hydrolysis (Figure 2A,B). The RD comprises 19 predicted protein kinase A (PKA) phosphorylation sites, six of which are phosphorylated in vivo and regulate CFTR activity. The amino-terminus contains a lasso motif that anchors the protein to the cell membrane, while the carboxy-terminus features a PDZ-interacting domain that connects CFTR to the cytoskeleton (Figure 2A,B) [15,16].

The MSDs consist of six transmembrane segments (TM1–6 and TM7–12) which are individually joined by four intracellular loops (ICL1–2 and ICL3–4), extending into the cytoplasm [17,18,19]. Notably, ICL2 interacts predominantly with NBD2, ICL4 interacts with NBD1, and ICLs 1 and 3 interact with both NBDs, forming dynamic interfaces critical for protein folding, trafficking, and function. The TMs of each MSD are also joined by short extracellular loops (ECLs 1–3 and ECLs 4–6) that are exposed to the extracellular side of the cell. The NBDs contain catalytic subdomains with Walker A and B motifs as well as alpha-helical subdomains housing the conserved ABC signature motif [20,21]. Remarkably, NBD1 contains two non-conserved regions: the regulatory insert (RI) near the N-terminus and the regulatory extension (RE) near the C-terminus. The RI is disordered in NBD1 crystal structures and is involved in the regulation of CFTR gating but it is not essential for protein trafficking [20,21]. It has been suggested that in addition to affecting CFTR phosphorylation-dependent gating, RI movements may also modulate the interactions between ICL1 and NBD1 [22]. Recent cryo-electron microscopy (cryo-EM) studies have provided high-resolution (3.9 Å) insights into CFTR, revealing positively charged residues lining the ion conduction pathway and identifying key amino acids interacting with the intracellular and extracellular regions of the pore. Comparative structural analyses between CFTR and other ABC proteins have also unveiled unique differences in regions responsible for ion conduction and channel gating. These distinct structural features might represent potential therapeutic targets for small molecules aimed at modulating defective CFTR ion transport or expression [23,24,25].

## 4. CFTR Gating, Folding, and Trafficking

CFTR channels open and close in bursts, with each burst consisting of multiple open and closed states. The transition from a closed to an open state is driven by substantial conformational changes involving both the NBDs and the MSDs. In its dephosphorylated state, the RD inhibits NBD dimerization, maintaining the channel in a closed conformation. Phosphorylation of the RD by cyclic AMP-dependent protein kinase (PKA), induces structural rearrangements that allow ATP binding and hydrolysis at the NBDs. These processes drive the opening and closing of the channel, enabling chloride ions to flow across the membrane. The hydrolysis of ATP triggers the dissociation of the NBDs, leading to channel closure. This tightly regulated gating mechanism is fundamental for CFTR’s role as a chloride channel [23,26]. Some mutations in the CFTR gene interfere with or disrupt the gating process, severely impairing CFTR channel function. To counteract these defects, potentiators and activators have been developed. Potentiators enhance chloride ion transport by increasing the open probability (P0) of mutant CFTR proteins at the PM (Figure 3A), while activators are small molecules that do not bind directly to CFTR but activate the channel by regulating the intracellular levels of cAMP or ATP (Figure 3B).

CFTR biogenesis starts with its co-translational insertion into the endoplasmic reticulum (ER) membrane, where it undergoes a complex folding process involving both co-translational and post-translational events [27,28,29]. Proper folding is crucial for CFTR function, but this process is particularly prone to errors due to mutations in the CFTR gene. The ER-associated degradation (ERAD) quality control system plays a key role in this context: it allows correctly folded proteins to exit the ER while directing unfolded or misfolded CFTR proteins toward degradation. Once correctly folded, CFTR is transported to the Golgi apparatus where it undergoes additional maturation and glycosylation steps before being trafficked to the PM. At the PM, CFTR is subjected to further quality control, undergoing cycles of endocytosis and recycling. Defective CFTR proteins are ubiquitinated and targeted for lysosomal degradation, reducing the number of functional channels available at the cell surface to mediate ion transport [27,28,29,30,31,32,33]. Correctors have been identified and developed to facilitate the trafficking of defective CFTR to the cell surface, though their efficacy varies across different mutations and patient populations (Figure 3B). In addition, stabilizer agents that increase surface stability and reduce endocytosis could play a crucial role in further enhancing CFTR functional expression, offering valuable therapeutic benefits for individuals with CF.

## 5. Current Strategies to Ameliorate Defective CFTR Function: Potentiators and Co-Potentiators

The development of CFTR potentiators has evolved over several decades. These molecules are currently categorized into two main types based on their mechanisms of action. Type 1 potentiators enhance CFTR gating by increasing the open probability (P0) of the channel. These small molecules are relatively well-characterized in terms of their mechanism of action and binding sites on the CFTR protein (Table 1). Type 2 potentiators, often referred to as co-potentiators, are individually less potent but significantly enhance the efficacy of type 1 potentiators when used in combination [34]. However, the precise mechanism of action for type 2 potentiators remains less clearly understood (Table 2). Despite considerable progress in potentiator development, no investigational or approved CFTR potentiators, including combinations of type 1 and type 2 potentiators, have been able to fully restore CFTR channel function to wild-type (WT) CFTR levels.

### 5.1. Potentiators

In the Nineties, researchers discovered that the natural isoflavone genistein could enhance CFTR gating through mechanisms independent of ATP binding to the CFTR protein [35]. While the exact binding sites of genistein remain unknown, it is hypothesized that this tyrosine kinase inhibitor interacts directly with CFTR at two distinct sites. One of these is a high-affinity site that delays the closing of the channel, thereby prolonging its open state. The other is thought to be a low-affinity region near the ATP binding sites, where genistein may inhibit ATP binding, reducing the probability of channel opening [36,37]. Despite its initial promise, clinical trials failed to demonstrate genistein’s efficacy as a CFTR potentiator (NCT00590538 and NCT00016744), prompting researchers to seek more effective compounds.

Further efforts to address the functional defect of CFTR led Verkman’s group to identify two compounds, phenylglycine (PG-01) and sulfonamide (SF-01), through high-throughput screening (HTS) assays of an initial pool of 50,000 molecules [38]. These compounds showed some ability to potentiate the gating of the F508del CFTR mutant [38,39]. However, clinical testing of these compounds was discontinued due to rapid metabolism in vivo and limited efficacy on non-F508del mutants. Despite this, their identification marked a significant milestone in CFTR modulator research. Notably, SF-01 shares structural and chemical similarities with ivacaftor (VX-770), the first approved CFTR potentiator. Additionally, P5, another compound from Verkman’s early set, served as a lead compound for the development of the investigational potentiator GLPG-1837 [40].

Around the same time, Vertex Pharmaceuticals conducted an HTS of a library of 228,000 compounds that led to the identification of four scaffolds with potentiator activity [41]. The most promising lead compound from this screen, VRT-484, was subjected to a series of medicinal chemistry optimizations, culminating in the development of the derivative VRT-715. While VRT-715 exhibited robust potentiator activity, it faced significant solubility limitations in both aqueous and organic solvents, hindering its therapeutic potential. To overcome these challenges, further structural modifications were undertaken, ultimately leading to the discovery of ivacaftor (VX-770) [42,43,44].

#### 5.1.1. Ivacaftor (VX-770)

In January 2012, ivacaftor (VX-770), marketed under the name Kalydeco, was approved by the U.S. Food and Drug Administration (FDA) for the treatment of patients with the G551D mutation. Subsequently, in July 2012, Kalydeco received marketing authorization valid across the European Union (EU). Its success in improving CFTR-mediated chloride transport across the cell membrane prompted rapid extension of its approval to eight additional Class III mutations (G1244E, G1349D, G178R, G551S, S1251N, S1255P, S549N, S549R), as well as the Class IV R117H mutation. Currently, Ivacaftor is authorized for use in CF patients aged one month or older who have at least one mutation that causes CF [43,45,46,47]. In the same year of its approval, Eckford and colleagues provided the first insights into ivacaftor’s mechanism of action, employing a reconstitution system with purified CFTR protein to elucidate its effects. They demonstrated that ivacaftor directly binds to both WT and mutant G551D or F508del CFTR proteins, at an allosteric site distinct from the canonical catalytic site. This binding mediates CFTR channel potentiation in a non-conventional, ATP-independent manner [48]. Jih and Hwang further explored this mechanism using electrophysiological recordings. Their results showed that ivacaftor enhances CFTR function by decoupling gating cycles from ATP hydrolysis. Their data also suggested that ivacaftor’s binding site was most likely located on the MSDs rather than on the RD [49].

In subsequent years, significant research efforts were made to determine ivacaftor’s binding site. Initially, NBD2 was suggested as the binding site; however, this hypothesis was later dismissed when VX-770 was found to potentiate CFTR even in the absence of this domain [50]. Byrnes and colleagues utilized a hydrogen/deuterium exchange assay coupled with mass spectrometry to probe CFTR conformational changes and interactions with VX-770. Their findings identified ICL4 as the binding region, as this site was protected from hydrogen–deuterium exchange in the presence of VX-770 [51]. To better elucidate the molecular interactions between ivacaftor and CFTR, Liu and co-workers determined a cryo-EM structure of ivacaftor in complex with phosphorylated E1371Q CFTR in the presence of saturating ATP-Mg^2+^ (10 mM) at an overall resolution of 3.3 Å (PDB: 6O2P). Their findings demonstrated that ivacaftor binds to CFTR at the protein–lipid interface, specifically docking into a cleft formed by TM helices 4, 5, and 8 [52]. In a subsequent study, Csanady and Torocsik explored the solubility and potency of VX-770 by analyzing the activation of wild-type (WT) and mutant CFTR channels in cell-free membrane patches. Their experiments demonstrated that the aqueous solubility of VX-770 is two orders of magnitude lower than that previously reported by Van Goor and colleagues [53]. From these findings, they proposed a kinetic model in which two VX-770 molecules bind to two independent sites on CFTR with equal affinity, working cooperatively to potentiate the channel [54]. In 2020, Righetti and collaborators used X-ray crystallographic data from Liu and colleagues [52] to explore the docking mode of VX-770 within human CFTR. Through an integrative approach combining molecular docking, pharmacophore mapping, and quantitative structure–activity relationship (SAR) analysis, they identified residues F236, Y304, F305, S308, A309, and F312 in MSD1 and F931 in MSD2, which are located in a crevice within the MSDs, as critical for the proper positioning of VX-770 in the MSDs [55]. Subsequently, Laselva and collaborators employed photoactivatable ivacaftor probes to map VX-770’s binding sites on the CFTR protein. Their results identified two primary regions of interaction: one within ICL4 at the NBD1:MSD2 interface and another at the MSD1/MSD2 and membrane lipid interface, corroborating findings from previous cryo-EM studies [56]. Levring and colleagues explored ivacaftor’s effect on CFTR gating through a combination of ensemble functional measurements, single-molecule fluorescence resonance energy transfer (FRET), electrophysiology, and kinetic simulations. They found that the two NBDs of human CFTR dimerize prior to channel opening. In this model, VX-770 is thought to enhance channel activity through an allosteric gating mechanism, where ATP hydrolysis-induced conformational changes within the NBD-dimerized channel regulate chloride conductance. Notably, gating mutations such as G551D should reduce the efficiency of NBD dimerization, while ivacaftor should enhance channel activity by promoting pore opening while stabilizing the NBD dimer state [57]. Recently, Ersoy and collaborators performed computational analysis showing that certain residues in ivacaftor’s binding site are allosteric sources, resembling those at the ATP binding site. Their data suggested that ivacaftor acts as an allosteric modulator, enhancing CFTR PO by mimicking the allosteric signaling pathway triggered by ATP binding [58].

#### 5.1.2. Other Investigational Potentiators

A deuterated derivative of ivacaftor, known as deutivacaftor (VX-561, CTP-656), has demonstrated enhanced stability compared to its parent compound [59]. This optimized molecule is currently being evaluated in clinical trials in combination with two correctors, vanzacaftor (VX-121) and tezacaftor (VX-661), to assess its safety and efficacy in adults with CF carrying CFTR gating mutations (NCT05033080, NCT05076149).

ABBV-974 (also known as GLPG-1837) and ABBV-2451 (GLPG-2451) were collaboratively developed by Galapagos NV and AbbVie Pharmaceuticals in 2017 and 2021, respectively. ABBV-974, a derivative of compound P5 initially described by Verkman and collaborators, exhibited nearly threefold higher efficacy in potentiating the gating of mutant G551D compared to VX-770, albeit it showed lower potency [60]. GLPG-1837 appears to target the same binding site as VX-770, located at the interface between MSD1 and MSD2, which explains the lack of additive effect when the two compounds are used together [61,62]. Importantly, GLPG-1837 does not seem to interfere with the activity of CFTR correctors, making it suitable for inclusion in combination therapies [63,64,65]. To address pharmacokinetic and potency limitations associated with GLPG-1837, the derivative compound ABBV-2451 was developed [66,67]. Despite its design improvements, this derivative failed to demonstrate sufficient clinical efficacy, ultimately halting its further development. Consequently, research efforts shifted toward more promising candidates in the pursuit of effective CFTR potentiators.

Another notable compound, the potentiator ABBV-3067 (navocaftor), was also co-developed by Galapagos NV and AbbVie Pharmaceuticals. This potentiator demonstrated significant activity in primary cells derived from patients harboring class III CFTR mutations. Currently, the FALCON clinical trial (NCT0396988) is evaluating the safety, tolerability, and efficacy of GLPG-2451 in CF patients with class III CFTR mutations, in combination with the corrector GLPG-2222.

Dirocaftor, also known as PTI-808 is a CFTR potentiator developed by Proteostasis Therapeutics (now Yumanity Therapeutics). It bears structural similarities to VX-770. In Phase 2 studies (NCT03500263), dirocaftor was tested in combination with posenacaftor (PTI-801) and nesolicaftor (PTI-428), demonstrating significant improvements in sweat chloride concentration and lung function in CF patients carrying the F508del mutation [68]. Despite these promising outcomes, no further development has been planned for this molecule.

The potentiator icenticaftor, also known as QBW251, was initially investigated as a therapeutic candidate for the treatment of CF between 2014 and 2016 in a clinical trial sponsored by Novartis Pharmaceuticals (NCT02190604). However, this clinical study was discontinued due to the lack of significant efficacy in this group of patients [69]. Subsequently, Novartis redirected the development of icenticaftor towards other pulmonary diseases, including chronic obstructive pulmonary disease (COPD) and bronchiectasis, where it has shown more promising therapeutic potential [70].

In recent years, peptides have garnered attention as therapeutic candidates for respiratory diseases, including CF. These small molecules possess unique features such as high specificity, efficacy, and low toxicity, making them attractive for clinical applications. Traditionally, therapeutic peptides for CF encompassed antimicrobial peptides, protease inhibitors, mucus clearance agents, and wound healing promoters. These agents have demonstrated efficacy in mitigating CF-associated lung infections, improving mucus clearance, and promoting epithelial repair [71,72,73,74].

A more recent area of research has focused on the potential of peptides to modulate CFTR function directly. In 2021, Ferrera and collaborators identified Esc peptides, a class of antimicrobial peptides (AMPs) derived from frog skin as potent potentiators of CFTR. Using electrophysiological techniques and computational studies they demonstrated that Esc peptides enhance CFTR-mediated ion currents by directly interacting with the F508del-CFTR mutant. Beyond potentiating defective CFTR channels, Esc peptides exhibit dual function, eradicating lung infections, and promoting airway wound repair. These novel properties position Esc peptides as a promising candidate for the development of multimodal CF therapies [75].

In 2016, Faure and co-workers identified the CB subunit of crotoxin, a compound derived from the venom of *Crotalus durissus terrificus* as a novel ligand and allosteric modulator of CFTR. Using electrophysiological studies in *Xenopus laevis* oocytes, CFTR-expressing HeLa cells, and ex vivo mouse colon tissue, the authors demonstrated that CB interacts with the NBD1 domain of both WT and F508del CFTR. This interaction enhances chloride channel currents (potentiator effect), promotes the appearance of glycosylated CFTR, and facilitates the functional expression of F508del-CFTR at the plasma membrane following 24-h treatments (corrector effect). Molecular docking and hydrogen–deuterium exchange analyses revealed that the CB subunit binds to the ABCβ and F1-like ATP-binding subdomains of F508del NBD1 [76]. Building on this work, the authors utilized a structure-based in silico approach to design peptides mimicking the CB-F508del NBD1 interface. These peptides interact with the same region as the CB subunit and act as potentiators of phosphorylated F508del CFTR. Some peptides also exhibit additive effects when combined with VX-770, further supporting their therapeutic potential in CF, particularly for patients unresponsive to current CFTR modulators [77].

**Table 1 cimb-47-00119-t001:** Investigational and approved CFTR type 1 potentiators.

Name	Structure	References
Ivacaftor (VX-770) N-(2,4-ditert-butyl-5-hydroxyphenyl)-4-oxo-1H-quinoline-3-carboxamide	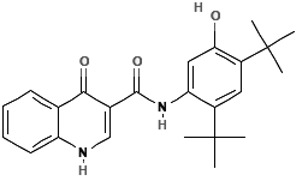	[42,43,44,45,46,47,48,49,50,51,52,53,54,55,56,57,58]
Deutivacaftor (VX-561, CTP-656) N-[2-tert-butyl-4-[1,1,1,3,3,3-hexadeuterio-2-(trideuteriomethyl)propan-2-yl]-5-hydroxyphenyl]-4-oxo-1H-quinoline-3-carboxamide	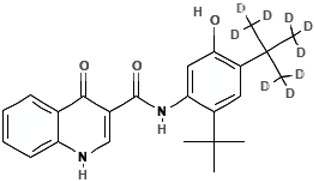	[59]
ABBV-974 (GLPG-1837) N-(3-carbamoyl-5,5,7,7-tetramethyl-4H-thieno[2,3-c]pyran-2-yl)-1H-pyrazole-5-carboxamide	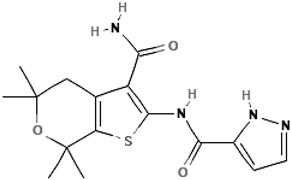	[50,60,61,62,63,64,65]
ABBV-2451 (GLPG-2451) 3-amino-N-[(2S)-2-hydroxypropyl]-5-[4-(trifluoromethoxy)phenyl]sulfonylpyridine-2-carboxamide	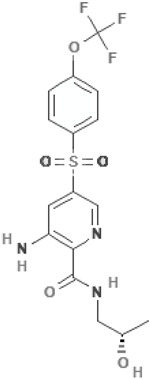	[66,67]
ABBV-3067 (Navocaftor) [5-[3-amino-5-[4-(trifluoromethoxy)phenyl]sulfonylpyridin-2-yl]-1,3,4-oxadiazol-2-yl]methanol	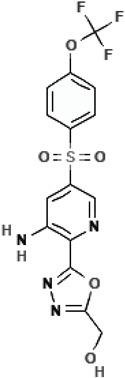	[67]
Dirocaftor (PTI-808) N-[5-hydroxy-2,4-bis(trimethylsilyl)phenyl]-4-oxo-1H-quinoline-3-carboxamide	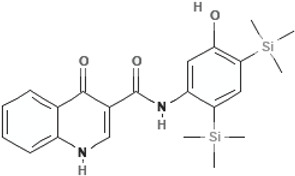	[68]
Icenticaftor (QBW251) 3-amino-6-methoxy-N-[(2S)-3,3,3-trifluoro-2-hydroxy-2-methylpropyl]-5-(trifluoromethyl)pyridine-2-carboxamide	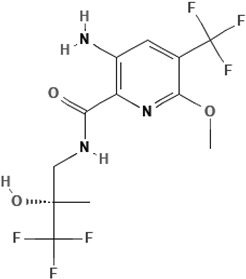	[69,70]
Esc Peptides	N.A.	[75]

### 5.2. Co-Potentiators

In 2019, Puay-Wah Phuan and his team introduced the concept of co-potentiator therapy for CF. This approach was specifically designed to address CFTR mutations that show insufficient response to conventional treatments involving standalone potentiators or correctors [34]. To identify molecules with co-potentiator activity that could synergize with ivacaftor, several research groups conducted HTS on drug and natural product libraries (Table 2). Among the co-potentiators discovered, the flavone apigenin demonstrated the ability to enhance the function of the W1282X CFTR mutation when combined with ivacaftor and forskolin [40]. However, the exact mechanism by which apigenin exerts its modulatory effect on CFTR remains largely unclear. Further investigations led to the identification of ASP-11, an arylsulfonamide-pyrrolopyridine compound, which demonstrated a synergy effect with VX-770 and forskolin, in potentiating the function of CFTR carrying mutations such as N1303K as well as other NBD2 mutations like I1234del, W1282X, and Q1313X. Notably, ASP-11 did not show any additive or synergistic effects in enhancing the function of the F508del mutation, suggesting that its efficacy may be mutation-specific and potentially tied to particular functional domains of the CFTR protein [78]. Phuan and colleagues also identified a series of spiro[piperidine-4,1-pyrido[3,4-b]indole] compounds with co-potentiator properties through HTS studies [78]. Among the 37 analogs identified, compound 2i emerged as the most effective in co-activating the N1303K CFTR mutant [79].

In another study, Liu and colleagues performed electrophysiological analyses to assess the efficacy of the investigational co-potentiator CP-628006. Their results showed that CP-628006, when used in combination with ivacaftor, significantly enhanced CFTR function in epithelial cells derived from human bronchial epithelia (HBE). The compound exhibited a particularly strong effect on F508del CFTR compared to the G551D CFTR gating mutation. Notably, unlike ivacaftor, which modulates CFTR independently of ATP, CP-628006’s effect on G551D-CFTR was found to be ATP-dependent. This observation suggests that CP-628006 may interact directly with the ATP-binding site of NBDs [80].

Rab and colleagues performed an extensive screening of 300,000 drug candidates to assess their potential to enhance CFTR trans-epithelial transport, either alone or in combination with VX-770. Following this screening, a structure–activity relationship (SAR) campaign identified a novel class of 7H-[1,2,4]triazolo[3,4-b][1,3,4]thiadiazines, which exhibited a substantial impact on CFTR activation when combined with VX-770. Among these molecules, the compound referred to as hit compound 3, demonstrated remarkable efficacy, increasing the activity of G551D mutant CFTR channels by approximately 400% compared to VX-770 alone. This enhancement effectively restored CFTR function to levels approaching those of WT CFTR in the Fischer rat thyroid (FRT) model system [81].

In a 2024 study, Liu and collaborators combined molecular docking, electrophysiology, cryo-EM, and medicinal chemistry to identify novel CFTR modulators. Using the CFTR structure in a complex with ivacaftor, approximately 155 million molecules were virtually screened against the potentiator-binding site of CFTR. From this extensive pool, 53 compounds were synthesized and optimized through structure-based methods. This effort led to the discovery of a new potentiator scaffold, pyrrolo-quinazolines, which exhibited mid-nanomolar affinity and was structurally distinct from previously known CFTR potentiators. One of the compounds of this novel potentiator class, I1421 demonstrated favorable physical and pharmacokinetic properties. It was easily formulated for both oral and intravenous delivery in mouse models, exhibiting significant bioavailability. These promising attributes suggest that the I1421 scaffold holds potential for further development and optimization as effective CFTR co-potentiators. Interestingly, this study also identified compounds that acted as inhibitors by binding to the same potentiator site, revealing that this membrane-exposed allosteric site could be targeted to either enhance or inhibit CFTR activity [82].

In another recent study, Spallarossa and collaborators applied sequential chemical synthesis approaches to develop mono- and diacyl thiourea derivatives, aimed at rescuing F508del-CFTR function. The compounds exhibited a favorable safety profile in both tumoral and normal cell lines. Some of them, including 6d, 6f, and especially 6i, displayed significant additive effects when combined with VX-809. These results suggest that thiourea derivatives may complement existing modulators, offering a promising opportunity for synergistic pharmacological approaches to enhance F508del-CFTR processing and trafficking [83].

**Table 2 cimb-47-00119-t002:** Investigational co-potentiators or type 2 potentiators.

Name	Structure	References
Apigenin 5,7-Dihydroxy-2-(4-hydroxyphenyl)-4H-chromen-4-one	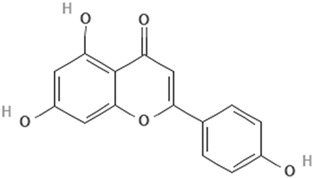	[40]
Compound 2i 6′-Methoxy-1-(2,4,5-trifluorobenzyl)-2′,3′,4′,9′-tetrahydrospiro[piperidine-4,1′-pyrido[3,4-b]indole]	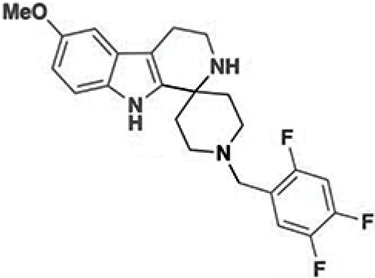	[79]
CP-628006 (4bS,7S,8aR)-4b-benzyl-7-hydroxy-N-[(2-methylpyridin-3-yl)methyl]-7-(3,3,3-trifluoropropyl)-5,6,8,8a,9,10-hexahydrophenanthrene-2-carboxamide	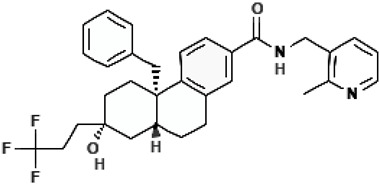	[80]
I1421 (S)-1-(3-Amino-6-(hydroxymethyl)-1H-indazol-1-yl)-3-(4-fluorphenoxy)-2-methylpropan-1-one	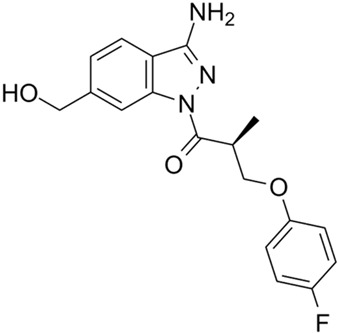	[82]
6i (E)-1-(3-Benzoyl-2-thioxotetrahydropyrimidin-1(2H)-yl)-3-phenylprop-2-en-1-one	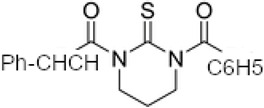	[83]

## 6. Strategies Aimed at Increasing the Amount of CFTR at the Plasma Membrane: Correctors

As already mentioned, correctors are small molecules designed to address CFTR mutations characterized by processing defects, such as F508del. These compounds directly bind to mutant CFTR, aiding its proper folding within the endoplasmic reticulum (ER) and facilitating its maturation in the Golgi apparatus. This process enhances the transport of CFTR to the plasma membrane (PM) while concurrently reducing its degradation. By increasing the availability of CFTR channels at the cell surface, correctors lay the groundwork for potentiators, which subsequently enhance channel activity, working in synergy to restore CFTR function and mitigate the defects associated with CF.

Correctors have been classically categorized into three main types based on their mechanisms of action, each targeting distinct aspects of CFTR folding and stability [84]. Type 1 correctors such as lumacaftor (VX-809) and tezacaftor (VX-661) stabilize the interface between NBD1 and MSDs. By binding to hydrophobic pockets within MSD1, they facilitate CFTR early folding stages reducing premature degradation and increasing the likelihood of successful trafficking to the PM (Table 3). Type 2 correctors, exemplified by compounds like corr4a, target NBD2 and its interactions with MSD interfaces. These correctors improve the assembly of CFTR domains, strengthening the protein’s structural integrity and facilitating its maturation and transport to the PM (Table 4). Finally, Type 3 correctors are designed to directly stabilize NBD1 and promote its interactions with other CFTR domains (Table 5).

The understanding that individual correctors used individually are often not sufficient to fully restore CFTR expression has driven the development of combination therapies aimed at achieving synergistic correction of CFTR defects. These strategies utilize the complementary action of two or more correctors, each targeting specific domains or different folding defects within the CFTR protein, to achieve a more comprehensive rescue of CFTR processing. A notable example is Trikafta, which combines the correctors elexacaftor (VX-445) and tezacaftor (VX-661) with the potentiator ivacaftor (VX-770). This therapeutic regimen has proven remarkable efficacy in addressing both CFTR folding and functional defects, leading to clinically significant improvements in CFTR activity and pulmonary function in patients with at least one F508del mutation.

### 6.1. Type 1 Correctors

One of the earliest milestones in the search for CFTR correctors was the identification of VRT-325, a quinazoline compound discovered using HTS. VRT-325 demonstrated effectiveness in enhancing the expression of the mature isoform of F508del CFTR at the PM [41]. However, concerns about its specificity soon emerged, as VRT-325 was found to rescue not only CFTR but also P-glycoprotein, another ATP-binding cassette (ABC) transporter closely related to CFTR [85]. In addition, higher concentrations of VRT-325 were reported to exert inhibitory effects, reducing the chloride current essential for CFTR function [86]. These findings raised questions about the therapeutic applicability of VRT-325 and prompted further investigations into its mechanism of action as well as its binding sites on the CFTR protein. Initial studies suggested that NBD1 was the primary target of VRT325 [87]. However, subsequent research indicated that VRT-325 facilitates the interaction between NBD1 and the two MSDs, thereby improving their stability and cooperative function [88,89]. Amico and colleagues, employed truncation mutants to demonstrate that VRT-325 specifically increased the expression and stability of the N-terminal half of F508del-CFTR, which includes the MSD1 and NBD1 domains [90]. To improve its pharmacological profile and efficacy as a corrector, VRT-325 underwent extensive medicinal chemistry modifications, resulting in the development of VRT-768 [91]. This derivative demonstrated an improved ability to enhance CFTR maturation and ion channel function in Fischer rat thyroid (FRT) cells expressing the F508del mutation. However, despite these improvements, the efficacy of VRT-768 remained modest, prompting further optimization efforts. These efforts culminated in the development of lumacaftor (VX-809), which showed significantly higher efficacy in promoting the maturation and trafficking of F508del CFTR to the PM [91]. Lumacaftor’s success marked a turning point in CFTR modulator research, providing the foundation for combination therapies, such as Orkambi (VX-770 + VX-809), and paving the way for the development of next-generation correctors.

#### 6.1.1. Lumacaftor (VX-809)

As mentioned above, in 2015, VX-809, in combination with the potentiator ivacaftor (VX-770), became the first corrector approved by both the FDA and EMA for CF patients aged 12 and older carrying at least one copy of the F508del mutation [44]. The availability of this new drug specifically targeting the folding and maturation defects of the CFTR protein significantly broadened the therapeutic landscape for CF patients, particularly those with processing-deficient mutations, offering renewed hope and improved outcomes for this population.

Early research into the mechanism of action and binding sites of VX-809 suggested that this corrector directly interacts with F508del CFTR or proteins associated with its synthesis or degradation [91]. Initial silico studies by Okioneda and colleagues [84] pointed to the NBD1:ICL4 interface as a critical target site for VX-809. This hypothesis was supported by Farinha and co-workers [92], who showed that VX-809 facilitates the assembly of this interface, enabling proper CFTR maturation. However, subsequent investigations conducted by Loo and his collaborators using truncation mutants and isolated CFTR domain revealed that VX-809 primarily increases the stability of MSD1, suggesting that MSD1 may serve as the primary binding site of the corrector [93]. Further studies clarified that VX-809 does not act directly on the thermodynamic instability of NBD1, but instead stabilizes the MSD1 domain during the early stages of CFTR folding [94]. Additionally, Ren and co-workers proposed that VX-809 attenuates the defects in NBD1 through an allosteric mechanism, without directly targeting the NBD1:ICL4 interface [95]. Subsequent studies by Eckford and colleagues pointed out that VX-809 plays a dual role in CFTR correction. First, it aids in the co-translational folding of F508del-CFTR and, later, it binds to CFTR once it reaches the cell surface. This dual activity highlights VX-809’s ability to facilitate early folding while also stabilizing CFTR and enhancing its functional activity at the PM [96]. In 2015, Sinha and colleagues employed click chemistry approaches to definitively demonstrate that VX-809 directly binds to MSD1 [97]. Although some later studies continued to suggest alternative binding sites, such as NBD1 [98] or the NBD1:ICL1 interface [99], increasing evidence, from 2015 onwards, confirmed that MSD1 serves as the primary binding site for VX-809.

Further research efforts focused on investigating the impact of VX-809 on CFTR folding and stability. Uliyakina and co-workers examined the effects of VX-809 in combination with the potentiator VX-770 on F508del CFTR mutants in which two regulatory regions of NBD1, the regulatory insertion (RI) and the regulatory extension (RE), were removed. Their findings demonstrated that VX-809 significantly increased the expression, stability, and plasma membrane localization of the ΔRI mutants. Interestingly, the removal of the RI region appeared to have a negative impact on the functional rescue of CFTR via VX-770, suggesting that RI may play a dual role in the action of these modulators [100]. Kleizen and co-workers further investigated the folding and assembly of CFTR using a combination of biosynthetic radiolabelling protease susceptibility assays and pulse-chase experiments with domain-specific antibodies. Their results revealed that MSD1 folds concomitantly with translation and that VX-809 supports this process. Additionally, VX-809 was shown to facilitate the assembly of ICL1 with the N-terminal subdomain of NBD1, promoting the stabilization of this inter-domain complex [101]. In a paper published in 2022, Fiedorczuk and Chen provided a conclusive description of the binding mechanism of VX-809 and other type 1 and type 1II correctors using cryo-EM structural analysis of the human CFTR protein in complex with these correctors. Their study showed that the VX-809 binds directly to a hydrophobic internal pocket within the MSD1 domain, further confirming MSD1 as its primary binding site (PDB:7SVR) [102].

#### 6.1.2. Tezacaftor (VX-661)

To address the limitations of lumacaftor in treating certain CF patients, including variable responses, limited efficacy on lung function, drug–drug interactions, adverse effects, and poor tolerability, extensive medicinal chemistry optimizations led to the development of the next-generation CFTR corrector tezacaftor (VX-661) [103]. In February 2018, tezacaftor received FDA approval in combination with the potentiator VX-770, marketed under the brand name Symdeko. In Europe, the combination was approved by EMA in October 2018 under the brand name Symkevi. Since October 2019, tezacaftor has also been included in a triple combination therapy alongside the corrector elexacaftor (VX-445) and the potentiator ivacaftor (VX-770), marketed as Trikafta in the United States and Kaftrio in the European Union, with EMA approval in August 2020. Both therapeutic regimens are indicated for CF patients aged two years and older who carry at least one F508del mutation in the CFTR gene [104].

As a chemical derivative of VX-809, tezacaftor is generally believed to share a similar mechanism of action and binding site of lumacaftor, primarily targeting the MSD1 domain to stabilize CFTR folding and trafficking. Consequently, studies focusing on the precise binding site and mechanism of action of tezacaftor remain relatively less prevalent than those of lumacaftor. Molinsky and colleagues performed molecular docking analyses, using a full-length open-channel homology model of the CFTR protein and a closed-channel cryo-electron microscopy structure, to explore the binding sites of type 1 correctors VX-661, VX-809, and C18. Their findings, validated by immunoblotting, confirmed that these correctors share the same binding site and mechanism of action. Specifically, they stabilize the MSD1 domain by targeting a multi-domain pocket near residues F374 and L375 [105]. In 2023, Bongiorno and colleagues extended this analysis by investigating the binding sites of several CFTR correctors, including VX-809, VX-661, Corr4a, and VX-445, using biochemical and functional assays [106]. Their results showed that the correctors under investigation have specific effects on the expression and stability of different regions within the F508del CFTR protein. Specifically, VX-809 and its derivative VX-661 were confirmed to target and stabilize the MSD1 domain, consistent with findings from cryo-EM analyses conducted by Fiedorczuk and Chen [102]. These results, further validating the importance of MSD1 as a therapeutic target for correctors, underscored once more the critical role of this region in the folding, trafficking, and stabilization of CFTR.

#### 6.1.3. Other Type 1 Correctors

C18 also known as VRT-534 was developed through structure–activity relationship (SAR) analysis and structural optimization of VRT-768 [84]. This compound has been shown to bind directly to phosphorylated and reconstituted F508del-CFTR, increasing the levels of both core-glycosylated and complex-glycosylated forms of the mutant CFTR protein. C18 also promoted proper folding and processing of CFTR in airway primary cells derived from CF patients. In addition, treatment with C18 leads to a modest increase in protease resistance of full-length F508del-CFTR, suggesting a role in improving protein stability at the cell surface [96]. Interestingly, C18 exhibited a dual function, acting not only as a corrector but also as a potentiator, addressing both trafficking and gating defects of CFTR caused by CF mutations. This dual activity suggests that its action addresses either intracellular trafficking and stabilization at the PM of defective CFTR, enabling its proper activation and function [107].

In a 2018 study, Wang and colleagues reported the identification of ABBV-2222 (GLPG-2222), a novel CFTR corrector developed by AbbVie, which exhibited higher potency than VX-809 while maintaining comparable efficacy in restoring cell surface expression of F508del CFTR in patient-derived cells [108]. To investigate its mechanism of action, Singh and his colleagues used two suppressor mutants, R1S (harboring mutations G550E, R553Q, R555K, and F494N that stabilize the NBD1 domain) and R1070W (stabilizing the NBD1:MSD2 interface). Their biochemical assays revealed that ABBV-2222 was more effective in promoting the maturation of the CFTR protein (formation of band C) in baby hamster kidney (BHK) cells expressing the R1070W suppressor mutant than in cells expressing the R1S mutants. These findings suggested that ABBV-2222 primarily exerts its action by stabilizing the interface between NBD1 and MSD2 rather than directly addressing the structural defects in NBD1 itself. Consequently, ABBV-2222 was classified as a type C1 corrector [109]. Of note, ABBV-2222 demonstrated specificity for F508del CFTR, as it selectively corrected the processing defects of this mutant but failed to rescue the folding and trafficking defects of other misfolded proteins such as P-glycoprotein (G268V-PgP) or the hERG K+ channel (G601S-hERG) [109]. Despite its efficacy in correcting F508del CFTR expression in vitro and its good tolerability in patients participating in clinical trials [110], ABBV-2222 did not produce significant improvements in lung function. To further explore its potential, a Phase 2 clinical trial (NCT03969888) was conducted to evaluate its safety, tolerability, and efficacy in combination with the potentiator ABBV-3067 in adults with cystic fibrosis who were homozygous for the F508del mutation. The study was completed on 9 June 2022 with a total of 78 participants enrolled from several countries. However, the detailed results of this clinical trial have not yet been published.

FDL-169, developed by Flatley Discovery Lab, is a CFTR corrector designed to increase the presence of the mature F508del CFTR protein on the cell surface, potentially improving its stability and function. Despite encouraging preclinical results, clinical development has encountered challenges. In particular, a Phase 2 clinical trial (NCT03756922) assessing the safety, pharmacokinetics, and pharmacodynamics of FDL-169 in adults homozygous for the F508del mutation was ultimately suspended, leaving its future clinical prospects uncertain.

ARN23765 is a CFTR corrector molecule developed as part of the Task Force for Cystic Fibrosis (TFCF) project, a strategic initiative funded by the Fondazione Italiana per la Ricerca in Fibrosi Cistica (FFC) in collaboration with the Istituto Italiano di Tecnologia (IIT). The project began with a HTS campaign that identified a promising hit compound. Subsequent iterative cycles of chemical synthesis and functional/biochemical evaluation in different cell systems, including bronchial epithelial cells from CF patients, led to the discovery of a lead compound capable of rescuing F508del-CFTR at low nanomolar concentrations. Through further optimization of the lead compound’s chemical structure, ARN23765 emerged as a highly effective corrector, exhibiting an EC_50_ value of 38 picomolar in bronchial epithelial cells from patients homozygous for F508del. This remarkable potency represents a 5000-fold increase compared to currently available correctors [111,112]. To investigate its mechanism of action, HS-YFP assays were conducted in FRT and CFBE41o− cells, testing ARN23765 both alone and in combination with other correctors, including 4172 (a type 1II corrector) and VX-809. The additive effects observed with 4172 and the lack of additivity with VX-809 indicated that ARN23765 shares a similar mechanism of action and binding site with VX-809, classifying it as a type 1 corrector. Given its promising preclinical profile, ARN23765 was licensed to Sionna Therapeutics, a US-based biotechnology company, in 2021 for further development. Building on the findings obtained by the TFCF team, Sionna Therapeutics advanced the compound’s optimization, ultimately resulting in the identification of SION-676, a variant of ARN23765 with enhanced pharmacological activity and improved drug-like properties.

The psoralen-related compound TMA (4,6,4′-trimethylangelicin) has attracted considerable interest as a potential CFTR modulator due to its anti-inflammatory properties and its ability to enhance CFTR functional expression. TMA is known to target NF-κB and reduce IL-8 expression, key inflammatory markers in CF. Concurrently, it demonstrated the ability to potentiate the cAMP/PKA-dependent activation of WT CFTR and partially restore F508del CFTR-dependent chloride secretion in both primary and secondary airway cells from patients homozygous for the F508del mutation [113]. Mechanistic studies by Laselva and co-workers demonstrated that TMA, like VX809, interacts with CFTR’s MSD1, stabilizing its structure and strengthening the interaction between NBD1 and ICL4. For these characteristics, it has been postulated that TMA exerts a dual role either as a corrector or a potentiator [114,115]. Acknowledging its anti-inflammatory effects and CFTR-modulatory potential, TMA received an Orphan Drug Designation for CF treatment by the European Medicines Agency (EMA) in 2013 (EU/3/13/1137) [116]. However, despite its promising pharmacological profile, concerns arose regarding TMA’s poor solubility, phototoxicity, and mutagenicity, limiting its clinical applicability [117]. To address these limitations, new generation TMA analogs with improved safety profiles and enhanced modulatory properties were developed. Among these derivatives, three non-mutagenic promising hit compounds were identified: IPEMA (4-isopropyl-6-ethyl-4′-methylangelicin), PEMA (4-propyl-6-ethyl-4′-methylangelicin), and IPDMA (4-isopropyl-6,4′-dimethylangelicin). Notably, IPEMA retained CFTR potentiation activity and efficiently rescued F508del-CFTR-dependent chloride efflux at nanomolar concentrations in FRT-YFP-F508del, HEK-293, and CF patient-derived bronchial epithelial cells [118]. Further analyses revealed that also these derivatives exerted a dual action as correctors and potentiators. Furthermore, like their progenitor TMA, they bind to the MSD1 domain and stabilize the NBD1:ICL4 interface. This action facilitates CFTR folding and trafficking while preventing premature protein degradation [119]. Building on these findings, a small library of new derivatives was synthesized and evaluated for drug-likeness, pharmacokinetics, and CFTR modulator efficacy. Several derivatives, including 4-PhDMA and pANDMA mesylate, demonstrated improved safety profile, higher bioavailability, and the ability to correct mutant CFTR expression in F508del CFTR-expressing CFBE41o– cells. Notably, these derivatives displayed synergistic effects with VX-809, highlighting their compatibility with combinatorial therapies and potential for multimodulator regimens [120].

The group, led by Millo, developed and characterized two series of CFTR correctors, referred to as hybrids, which were synthesized by combining the aminoarylthiazole (AAT) core with the benzodioxole carboxamide moiety of VX-809. These derivatives were subjected to several assays aimed at assessing their potential as F508del-CFTR correctors and elucidating their mechanism of action [121]. Among the synthesized compounds, three promising derivatives—2a, 7a, and 7m—were identified and further investigated using in silico modeling. Computational studies revealed that these hybrids interact with a putative binding region at the MSD1:NBD1 interface. Biochemical analyses confirmed these results and showed that the selected compounds affect the expression and stability of the F508del NBD1 domain. To further explore their functional efficacy, a YFP functional assay was conducted in both secondary airway epithelial cells derived from F508del-CFTR patients. Functional testing demonstrated that 2a, 7a, and especially 7m exhibited additive effects when used in combination with type 1 correctors like VX-661, and type 3 correctors such as VX-445 [122].

**Table 3 cimb-47-00119-t003:** Investigational and approved type 1 correctors.

Corrector	Structure	References
VRT-325 4-(Cyclohexyloxy)-2-(1-(4-((4-methoxyphenyl)sulfonyl)piperazin-1-yl)ethyl)quinazoline	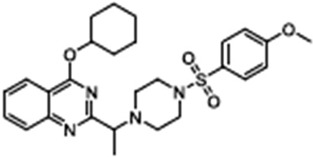	[41,85,86,87,88,89,90]
VRT-768 N-(2,4-diaminothiazol-5-yl)-4-[(2,4-dimethylphenyl)sulfonyl]benzamide	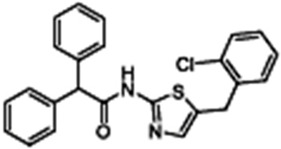	[91]
Lumacaftor (VX-809) 3-[6-[[1-(2,2-difluoro-1,3-benzodioxol-5-yl)cyclopropanecarbonyl]amino]-3-methylpyridin-2-yl]benzoic acid	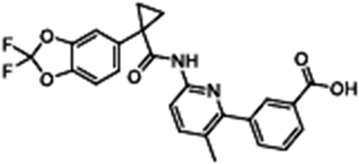	[44,84,92,93,94,95,96,97,98,99,100,101,102]
Tezacaftor (VX-661) 1-(2,2-difluoro-1,3-benzodioxol-5-yl)-N-[1-[(2R)-2,3-dihydroxypropyl]-6-fluoro-2-(1-hydroxy-2-methylpropan-2-yl)indol-5-yl]cyclopropane-1-carboxamide	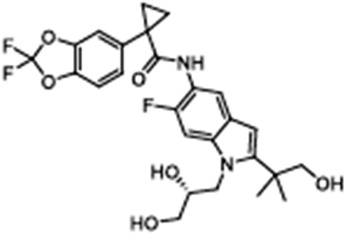	[103,104,105,106]
C18 (VRT-534) 1-(1,3-benzodioxol-5-yl)-N-[5-[(S)-(2-chlorophenyl)-[(3R)-3-hydroxypyrrolidin-1-yl]methyl]-1,3-thiazol-2-yl]cyclopropane-1-carboxamide	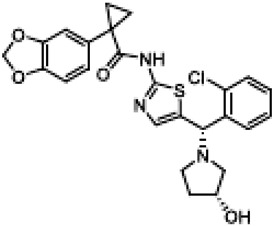	[84,96,107]
ABBV-2222 (GLPG-2222) 4-[(2R,4R)-4-[[1-(2,2-difluoro-1,3-benzodioxol-5-yl)cyclopropanecarbonyl]amino]-7-(difluoromethoxy)-3,4-dihydro-2H-chromen-2-yl]benzoic acid	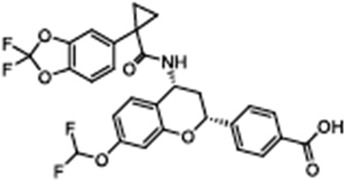	[108,109,110]
FDL-169 2-[7-ethoxy-4-(3-fluorophenyl)-1-oxophthalazin-2-yl]-N-methyl-N-(2-methyl-1,3-benzoxazol-6-yl)acetamide	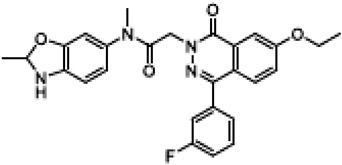	[Flatley Discovery Lab, clinical trial NCT03756922, 169]
ARN23765 4-{[(3-{[(2,2-difluoro-1,3-benzodioxol-5-yl)(methyl)carbamoyl]phenyl}{3-(trifluoromethyl)-4,5,6,7-tetrahydro-1H-indazol-7-yl]oxy}benzoic acid	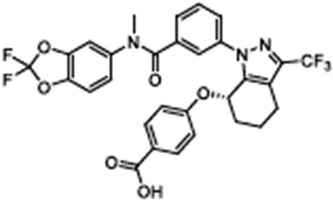	[111,112]
TMA (Trimethylangelicin) 4,6,9-trimethylfuro[2,3-h]chromen-2-one	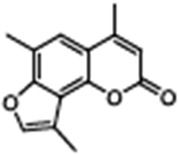	[113,114,115,116,117]
IPEMA 4-isopropyl-6-ethyl-4′-methylangelicin	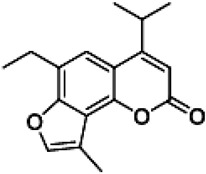	[118,119]
4-PhDMA 6,9-Dimethyl-4-phenyl-2H-furo[2,3-h]-1-benzopyran-2-one	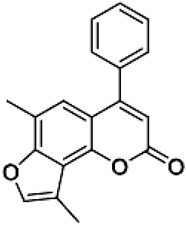	[120]
pANDMA mesylate 4,9-Dimethyl-6-(4′-aminophenyl)-2H-furo[2,3-h]-1-benzopyran-2-one	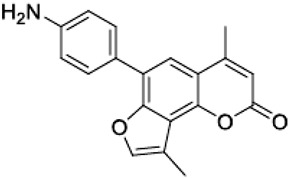	[120]
7m N-(5-([1,10-biphenyl]-4-carbonyl)-4-phenylthiazol-2-yl)-1 (benzo[d][1,3]dioxol-5-yl) cyclopropanecarboxamide	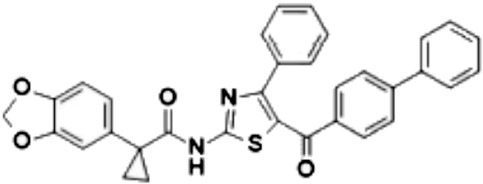	[121,122]

### 6.2. Type 2 Correctors

Type 1I correctors act via different mechanisms than type 1 correctors and have the potential to enhance their effect. As already mentioned, these molecules interact specifically with NBD2 and its interfaces promoting their stabilization. Although no type 1I corrector has yet received FDA approval, several investigational compounds have been identified and are under active study.

#### 6.2.1. Corr4a

One of the first correctors identified by HTS was corr4a, a bisaminomethylbithiazole compound. Corr4a demonstrated the ability to enhance the expression and trafficking of F508del CFTR in homozygous human bronchial epithelial cells. Its primary mode of action involved improving the folding and stability of the mutant CFTR protein during its maturation process, thereby facilitating its escape from endoplasmic reticulum-associated degradation (ERAD). Early studies also indicated that corr4a could increase CFTR-dependent chloride secretion, although its efficacy in functional rescue was modest compared to later-generation correctors [123]. Initially, the exact binding site of corr4a was not fully understood, but it was postulated that it interacts directly with the CFTR protein, targeting specific inter-domain interfaces and stabilizing them [89]. Later, Groove and colleagues employed biochemical approaches to identify MSD2 as the critical domain required for corr4a-mediated rescue of both expression and stabilization of the entire F5098del CFTR protein [124]. Subsequent studies by Laselva and colleagues supported these conclusions, confirming that corr4a interacts with MSD2 [115]. However, contrary evidence was presented by Okiyoneda and co-workers, who proposed that corr4a targets NBD2 rather than MSD2. Their biochemical studies suggested that corr4a stabilizes this nucleotide-binding domain, promoting its proper folding and integration into the CFTR protein structure [84]. Further supporting this hypothesis, Amico and collaborators used truncation mutants to definitively demonstrate that NBD2 is the primary binding site for corr4a. Their biochemical assays revealed that corr4a stabilizes isolated NBD2 when heterologously expressed in HEK-t cells, prolonging its half-life and suggesting a direct stabilizing effect on this domain [90]. Despite CORR-4a exhibiting limited potency, its discovery inspired the synthesis of more potent and clinically viable correctors, such as VX-809 and VX-661, which have since revolutionized CF treatment.

#### 6.2.2. Other Type 2 Correctors

VX-152 and VX-440 also known as olacaftor, are next-generation CFTR correctors developed by Vertex Pharmaceuticals. The properties of these correctors were evaluated as part of combination therapies for cystic fibrosis patients who were either homozygous for the F508del mutation or heterozygous for F508del with minimal function mutations in Phase 2 clinical trials (NCT02951182 for VX-440 and NCT02951195 for VX-152). However, based on the results obtained, the development of VX-152 and VX-440 was ultimately discontinued as Vertex redirected its focus on the development of VX-445 (elexacaftor) and VX659 (bamocaftor), which exhibited superior pharmacological properties and enhanced efficacy and better long-term safety profiles [125].

ABBV-3221 (GLPG-3221) and ABBV-2737 (GLPG-2737) were identified through HTS and co-developed by Galapagos and AbbVie as type 2 correctors targeting specific defects in CFTR folding and trafficking. The preclinical evaluation highlighted that ABBV-3221 exhibited a favorable safety and pharmacokinetic profile. In vitro studies demonstrated that ABBV-3221, when used in triple combination therapies with a type 1 corrector and a potentiator, produced greater improvements in CFTR functional expression than the dual combination of the potentiator GLPG-1837 and the type 1 corrector ABBV-2222. This robust performance provided a strong rationale for ABBV-3221’s advancement into clinical trials as part of multi-modulator regimens for the treatment of CF [126]. Similarly, ABBV-2737 demonstrated efficacy both as a monotherapy and in combination with other correctors, including VX-809, VX-661, and ABBV-2222 [127]. Notably, the potency of ABBV-2737 increases approximately 25-fold in the presence of ABBV-2222, suggesting that their synergistic effect arises from binding to different sites on the CFTR protein [128]. In the phase 2a PELICAN study (NCT03474042), ABBV-2737 demonstrated significant improvements compared to placebo in CF patients homozygous for F508del who were already receiving lumacaftor/ivacaftor, confirming its ability to enhance CFTR function in combination therapy. Further clinical investigation in a Phase 1b study (NCT03540524) highlighted the limited efficacy of this compound on F508del CFTR gating but demonstrated its ability to inhibit WT CFTR in a dose-dependent manner. This unexpected property opens potential therapeutic applications beyond cystic fibrosis. For instance, CFTR inhibition could reduce cyst growth and kidney enlargement in patients with autosomal dominant polycystic kidney disease (ADPKD), suggesting that ABBV-2737 may be repurposed for other diseases where CFTR inhibition is advantageous.

The thiazole compound 4-(3-chlorophenyl)-N-(3-(methylthio)phenyl)thiazol-2-amine (FCG) was initially identified through structure–activity relationship (SAR) studies and proved to be a promising type 2 corrector for the treatment of cystic fibrosis (CF). Although FCG has a weaker potency compared to corr4a, it showed unique properties and significant CFTR corrector activity, including the ability to act synergistically with VX-809 or VX-661, warranting further exploration [129]. Early molecular docking studies suggested that FCG might bind within a pocket of the NBD1 domain of the CFTR protein. However, later biochemical analyses by Brandas and colleagues showed that FCG interacts with a region in NBD2, classifying this molecule as a type 2 corrector. Using HEK-t cells heterologously expressing different CFTR domains and domain combinations, the authors showed that FCG was able to increase the overall expression of F508del CFTR. In combination with VX-809, this effect was further enhanced, increasing both the overall expression and the maturation rate of the mutant protein. Molecular modeling analyses underpinned the results obtained [130].

In a recent 2024 study, Scanio and collaborators identified and explored the SAR of a novel series of 4-aminopyrrolidine-2-carboxylic acid derivatives as CFTR correctors. These compounds share structural similarities with pyrrolidine ether CFTR correctors, including the type 2 corrector ABBV-3221. Similarly to ABBV-3221, the newly synthesized derivatives were designed to function as part of a combination therapy involving type 1 correctors such as ABBV-2222 and potentiators like ABBV-3067, with the aim of enhancing the functional expression of defective CFTR proteins. Due to its favorable safety profile and pharmacokinetic (PK) properties in rats (including low CYP3A4 induction, low clearance, and a long half-life), combined with its acceptable potency and efficacy the 2-(dimethylamino)-3-pyridyl analog 15 emerged as a promising type 2 corrector to be used in modulator combination therapies aimed at targeting CFTR processing defects in CF [131].

**Table 4 cimb-47-00119-t004:** Investigational type 2 correctors.

Corrector	Structure	References
Corr-4 N-(2-(5-Chloro-2-methoxy-phenylamino)-4′-methyl-[4,5′]bithiazolyl-2′-yl)-benzamide, Corr-4, N-(2-(5-Chloro-2-methoxy-phenylamino)-4′-methyl-[4,5′]bithiazolyl-2′-yl)-benzamide	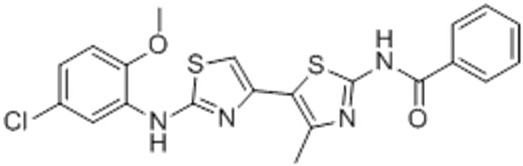	[84,89,90,115,123,124]
VX-440 Olacaftor (N-(benzenesulfonyl)-6-[3-fluoro-5-(2-methylpropoxy)phenyl]-2-[(4S)-2,2,4-trimethylpyrrolidin-1-yl]pyridine-3-carboxamide	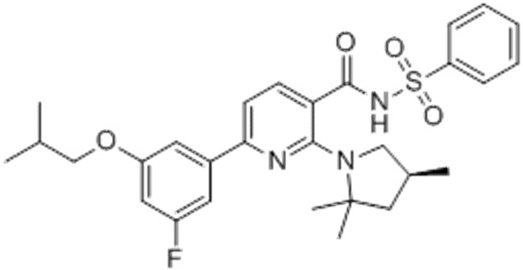	[Vertex Pharmaceuticals clinical trial NCT02951182]
ABBV-3221 (GLPG-3221) (2S,3R,4S,5S)-3-tert-Butyl-4-{[2-methoxy-5-(trifluoromethyl)pyridin-3-yl]methoxy}-5-(2-methylphenyl)-1-[(2S)-oxane-2-carbonyl]pyrrolidine-2-carboxylic acid	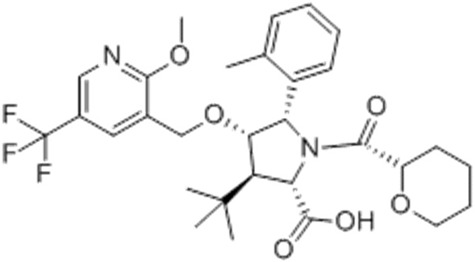	[126]
ABBV-2737 (GLPG-2737) 3-cyclobutyl-N-[(dimethylamino)sulfonyl]-1-(4-fluorophenyl)-4-[4-methoxy-1,4′-bipiperidin]-1′-yl]-1H-pyrazolo[3,4-b]pyridine-6-carboxamide	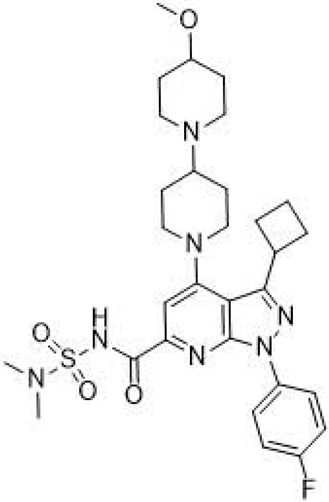	[127,128]
FCG 4-(3-chlorophenyl)-N-(3-(methylthio)phenyl)thiazol-2-amine	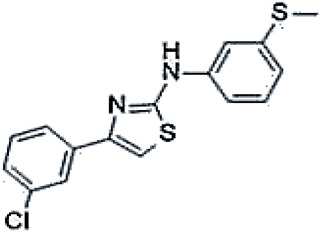	[129,130]

### 6.3. Type 3 Correctors

Given the modest therapeutic benefits achieved with currently approved CFTR modulators, additional HTS efforts have been launched to identify more potent and effective molecules capable of addressing the multifaceted defects of CFTR mutations. Concurrently, the focus has shifted toward the design and development of modulators suitable for inclusion in triple combination therapies. This strategic approach is built on the premise that synergistic interactions among modulators, each targeting specific structural defects within the CFTR protein, can amplify CFTR functional rescue and deliver therapeutic approaches with transformative outcomes. The initial type 3 correctors identified were 5-bromoindole-3-acetic acid (BIA) and its analog, 5-bromo-4-ethoxyindole-3-acetic acid (BEIA). However, these molecules were eventually discontinued due to the high effective concentrations required to achieve CFTR functional expression rescue [132].

Significant advancements in the identification of CFTR correctors were achieved by Veit and colleagues, who screened a library of approximately 600,000 compounds, leading to the isolation and characterization of novel type 1, 2, and 3 correctors [61]. While these compounds exhibited limited efficacy when used individually, their combinations with other CFTR modulators demonstrated a synergistic effect, significantly rescuing the functional expression of F508del CFTR in primary human bronchial epithelia (HBE) and nasal epithelial cells (HNE) derived from F508del homozygous patients. Among the identified compounds, pyrazole 4172 was classified as a type 3 corrector, being capable of directly binding to CFTR and stabilizing the NBD1 domain. Binding assays revealed that 4172 interacted with isolated F508del NBD1 with a binding of approximately 40 μM and exhibited an EC_50_ of around 3 μM in CFB41o- epithelial cells. Notably, 4172 demonstrated its greatest efficacy when used in combination therapies including type 1 and type 2 correctors [133].

#### 6.3.1. Elexacaftor (VX-445)

Following an extensive campaign involving HTS, structure–activity relationship (SAR) studies, and chemical optimization to enhance efficacy, pharmacokinetics and safety profiles of hit compounds, elexacaftor (VX-445) emerged as a potent third-generation corrector from Vertex Pharmaceuticals’ drug discovery pipeline. VX-445 demonstrated strong synergistic activity when combined with tezacaftor and ivacaftor, significantly improving the functional expression of the CFTR protein harbouring the F508del mutation. Initial phase 2 trials confirmed that the triple combination therapy exhibited favourable pharmacokinetics, pharmacodynamics and tolerability, highlighting its potential as a highly effective treatment for CF [134,135]. Subsequent Phase 3 clinical trials, TRAFFIC (NCT03525574) and TRANSPORT (NCT03525444), demonstrated that elexacaftor/tezacaftor/ivacaftor triple combination was effective in patients carrying at least one copy of the F508del mutation. Indeed, the administration of the therapy improved key indicators of lung function, including the forced expiratory volume in one second (FEV1), and significantly reduced pulmonary exacerbations and sweat chloride levels. Based on these positive outcomes, the triple combination therapy, marketed as Trikafta in the United States and Kaftrio in Europe, was approved by the FDA in 2019 and by the EMA in 2020 for use in patients aged 12 and older carrying at least one F508del mutation [136,137].

Given the transformative therapeutic potential of VX-445, numerous research groups have focused not only on evaluating the efficacy of VX-770/VX-661/VX-445 triple combination therapy in rescuing the functional expression of F508del CFTR and other class 2 processing mutations but also on elucidating the mechanism of action and identifying the binding site of VX-445 on CFTR. The path to understanding the mechanism of action of VX-445 began in 2020 with a study by Veit and colleagues that demonstrated that VX-445 synergistically enhances F508del CFTR processing in human bronchial epithelial cells when combined with type 1 or type 2 correctors. Since type 1 correctors target the NBD1-MSD1 interface and type 2 correctors interact with NBD2, the authors concluded that VX-445 acts as a type 3 corrector by directly stabilizing NBD1. Other pivotal results of this study showed that VX-445 binds directly to isolated F508del NBD1, preventing its unfolding and promoting CFTR stabilization. Furthermore, the study proved that the VX-445/VX-661/VX-770 triple combination significantly rescued several rare misprocessing mutations (S13F, R31C, G85E, E92K, V520F, M1101K, and N1303K) located across MSD1, MSD2, NBD1, and NBD2. These findings suggested an allosteric mechanism of correction, underscoring the potential of Trikafta not only to target F508del CFTR but also to address rare misfolding CFTR mutations, thereby broadening its therapeutic applicability [138]. Capurro and colleagues further investigated the efficacy and mechanistic characteristics of VX-445 in addressing the F508del mutation. Their study demonstrated that in primary bronchial epithelial cells derived from CF patients, VX-445, when combined with type 1correctors (VX-809, VX-661) and the type 2 corrector corr4a, restored approximately 60–70% of F508del CFTR function compared to non-CF cells. However, subsequent analyses, focusing on ubiquitination levels, resistance to thermal aggregation, protein half-life, and subcellular localization, revealed that VX-445 combined with VX-661, VX-770, or other type 1 and type 3 correctors did not fully normalize F508del-CFTR functional expression, which on the contrary retained a significant degree of instability and degradation susceptibility [139]. Becq and collaborators investigated the effects of the elexacaftor/tezacaftor double combination with or without ivacaftor, on the maturation, membrane localization, and function of F508del CFTR expressed in human primary airway epithelial cells and CFTR expressing cell lines (CFBE and BHK) [140]. Ussing chamber assays, whole-cell, patch-clamp recordings, Western blotting, and immune-localization assays, the authors demonstrated that the elexacaftor/tezacaftor/ivacaftor triple combination effectively rescued F508del CFTR maturation defects, promoted its apical membrane localization, and partially restored chloride channel function. Interestingly, their findings highlighted that the presence of ivacaftor in the triple combination limited the corrective effects observed with the double combination of VX-445 and VX-661 on F508del CFTR expression. This observation aligns with earlier studies, which also suggested that ivacaftor may interfere with the stabilization and trafficking of F508del CFTR, potentially counteracting the beneficial effects of the correctors in certain cellular contexts [138,139]. Continuing the exploration of the effect of the triple combination therapy on the F508del mutation, Stanke and co-workers analyzed the effects of elexacaftor/tezacaftor/ivacaftor on the expression and maturation of the Phe508del CFTR protein in rectal suction biopsies from 21 cystic fibrosis patients homozygous or compound heterozygous for the mutation. Western blot analyses revealed that treatment with elexacaftor/tezacaftor/ivacaftor significantly enhanced total CFTR protein levels in eight out of twelve patients. However, a substantial proportion of the rescued CFTR protein remained incompletely glycosylated, suggesting that, while Trikafta facilitates post-translational processing, it does not fully restore the mature, functional form of CFTR required for consistent, long-term clinical benefits [141]. In addition to these findings, Im and collaborators provided a comprehensive global profile of the CFTR folding process and assessed the impact of CFTR modulators lumacaftor and elexacaftor on the folding pathway of the Phe508del-CFTR variant. Their results indicated that elexacaftor increased the transport of CFTR to the Golgi apparatus, while the combination of lumacaftor and elexacaftor enhanced the overall domain assembly. However, these modulators were unable to fully correct the misfolding of the NBD1 domain, emphasizing the need for improved therapies targeting the persistent folding defects of this domain [142]. Soya and collaborators provided further insight into this aspect, examining the post-translational folding landscape of CFTR and other ATP-binding cassette (ABC) transporters using molecular dynamics simulations, biochemical techniques, and a hydrogen–deuterium exchange. Their findings revealed that mutations disrupting or stabilizing the NBD1-MSD1/2 interfaces significantly affect the post-translational folding of CFTR as well as other ABCC transporters, such as MRP1 (ABCC1) and ABCC6 in the endoplasmic reticulum. VX-809 and VX-445 were shown to bind allosterically or orthosterically to the MSD1/2 interface, facilitating the rescue of post-translational intermediates trapped in misfolded conformations, whether the mutation occurred in NBD1 or MSD1 [143]. Finally, Ersoy and collaborators employed computational analysis to investigate the allosteric impact of ivacaftor, tezacaftor and elexacaftor on the Phe508del-CFTR structure. Their results showed that while tezacaftor binds to regions with limited allosteric influence (termed “valleys”), the binding sites for ivacaftor and elexacaftor contain key allosteric residues (termed “primary allosteric sources”). These observations suggested that ivacaftor and elexacaftor play a significant role as allosteric modulators of CFTR [58].

Aware of the importance of a mutation-specific approach in optimizing therapy for CF patients, researchers have increasingly focused their studies on the characterization of the effect of VX-445, either alone or in combination with VX-770 and VX-661, on the biochemical and functional properties of rare CFTR mutations. In a 2021 study, Veit and colleagues explored the therapeutic potential of VX-445 for three specific CFTR missense mutations: P67L, L206W, and S549R. The study assessed both the biochemical and functional rescue of these CFTR mutants in bronchial epithelial cell lines and patient-derived human primary nasal epithelia (HNE). Retrieved results showed that VX-445, when combined with VX-661 or other type 1 correctors, significantly increased the plasma membrane density of P67L-, L206W-, and S549R CFTR mutants, providing a more effective correction compared to type 1 correctors alone. However, short-circuit current measurements in HNE revealed that VX-445 did not provide additional functional benefit beyond VX-661 alone for the S549R CFTR mutation. In contrast, for P67L and L206W, the combination of VX-445 and VX-661 resulted in a significantly greater functional rescue than VX-661 alone [144]. In the same year, Laselva and colleagues investigated the impact of Trikafta on two rare CFTR mutations, H609R and I1023_V1024del, both associated with severe lung disease. They found that the treatment with lumacaftor and ivacaftor did not significantly correct the functional expression of these mutations. However, the VX-770/VX-661/VX-445 triple combination therapy effectively rescued both the trafficking and function of H609R and I1023_V1024del, demonstrating the enhanced efficacy of Trikafta in addressing these difficult-to-treat mutations [145]. Laselva and colleagues extended their investigation to other class 2 mutations, including G85E, M1101K, and N1303K, for which Trikafta had not yet received approval. Their studies conducted in HNE cells derived from individuals carrying these rare CF-causing mutations revealed that the M1101K variant showed a favorable response to the triple combination, with significant improvements in both channel function and protein processing. In contrast, G85E and N1303K exhibited more limited responses. Notably, this study revealed an additional mechanism for VX-445: it appeared to act not only as a corrector but also as a potentiator [146].

The potentiator activity of VX-445 was the subject of investigation by other research groups. Veit and collaborators studied the effects of VX-445, both alone and in combination with VX-770, in F508del and gating mutations such as G551D and G1244E. Their studies showed that VX-445 significantly increased VX-770-potentiated currents, confirming that VX-445 functions as both a corrector and a co-potentiator [147]. Addressing the dual role of VX-445, Tomati and co-authors investigated the pharmacological properties of VX445 on the G1244E mutation, which causes a severe gating defect and is unresponsive to ivacaftor. Through molecular, biochemical, and functional analyses, they demonstrated that the cellular context strongly influenced the response of the G1244E CFTR mutant to VX445. Indeed, in heterologous expression systems, VX-445 primarily functioned as a co-potentiator, effectively rescuing the gating defect associated with the mutation. Conversely, in patient-derived nasal epithelial cells, VX-445 did not exhibit co-potentiation effects; rather, it enhanced the mature CFTR expression, likely by improving the stability of the mutant protein at the PM [148].

Continuing to study the effect of VX-445 on rare mutations, Hillenaar and colleagues investigated the early and late folding stages of A46D, G85E, R560S, G628R, L1335P, along with F508del, using biosynthetic radiolabeling and protease-susceptibility assays. Their results showed that most CFTR mutants exhibited biochemical responses specific to each modulator; however, these responses often did not correlate with the corresponding functional improvements in CFTR activity [149]. Using a deep mutational scanning approach, McKee and colleagues evaluated the impact of elexacaftor, tezacaftor, and their combination on the PM expression of 129 CF-causing mutations. They observed that elexacaftor and tezacaftor were most effective in correcting variants with defects near their respective binding sites, though most rescued variants displayed intermediate levels of PM expression compared to WT CFTR. When combined, the correctors synergistically enhanced the PM expression of CFTR variants distributed across different regions of CFTR, demonstrating the complementarity of their mechanisms of action [150]. Focusing on two specific CFTR variants, P67L and L206W, Kim and colleagues employed affinity purification-mass spectrometry to quantify changes in CFTR protein–protein interactions. VX-445 was shown to promote unique interactions with proteostasis factors involved in translation, folding, and degradation in a mutation-specific manner. Notably, siRNA-mediated knockdown of ribosomal subunit proteins increased the fully glycosylated form of P67L CFTR, sensitizing it to VX-445 and further enhancing its correction. These findings emphasize the role of translational dynamics in VX-445-mediated rescue [151]. Finally, Lefferts and colleagues studied Trikafta’s effects on 22 patient-derived intestinal organoids carrying rare CFTR variants not previously eligible for modulator therapy. Using the forskolin-induced swelling (FIS) assay, they observed significant restoration of CFTR function in 12 organoids representing 11 unique genotypes, supporting the potential of Trikafta for treating certain rare CFTR variants [152].

Concomitant with investigations into the mechanism of action and efficacy of VX-445 alone or in combination with F508del and other class 2 processing mutations, a major challenge for researchers has been the identification of its binding sites on the CFTR protein. Addressing this question required the application of advanced techniques, including cryo-electron microscopy (cryo-EM), molecular docking, and sophisticated biochemical approaches. These methodologies have provided critical insights into the structural interactions of VX-445 with CFTR, providing valuable information for the rational design of next-generation modulators. Using blind docking studies and molecular dynamics simulations of the 3D structures of MSD1 and NBD1, followed by biochemical experiments, Baatallah and collaborators identified two potential binding sites of VX-445 on CFTR: one unique on MSD1, and another shared with VX-809 on NBD1. The simultaneous binding of these two correctors to MSD1 was shown to enhance the allosteric interaction between MSD1 and NBD1, providing a plausible mechanistic explanation for the increased CFTR rescue efficacy observed when VX-445 is combined with other correctors [153]. As previously mentioned, cryo-electron microscopy has played a crucial role in providing high-resolution insights into the conformation of CFTR in the presence of modulators, revealing key interaction sites within its transmembrane and nucleotide-binding domains. In a landmark study, Fiedorczuk and Chen resolved the structure of F508del CFTR protein bound to different modulators, including elexacaftor alone (PDB: 8EIG), elexacaftor in combination with VX-809 (PDB: 8EIO), and with tezacaftor and lumacaftor (PDB: 8EIQ). Their findings revealed that elexacaftor occupies a shallow binding site within the plasma membrane, extending from the core of the lipid bilayer to the edge of the inner leaflet. The primary interaction occurs with TM helix 11 via electrostatic and van der Waals forces, while secondary interactions involve TM helices 2 and 10 and the lasso motif [102]. Wang and colleagues employed cryo-EM global conformational ensemble reconstruction to investigate the binding sites and dual activity of VX-445. Their study, focused on the G551D gating variant, showed that VX-445 binds to a cavity formed by TM helices 2, 10, and 11 along with the lasso motif. This interaction promoted the channel’s transition towards an open conformation, highlighting the potentiator role of VX-445. Furthermore, VX-445 binding was shown to stabilize NBD1 and the MSDs, underscoring its function as a corrector [154]. Bongiorno and co-workers employed biochemical approaches using isolated CFTR domains heterologously expressed in HEK-293 cells to identify the binding site and mechanism of action of VX-445. Confirming findings from cryo-EM studies, their research demonstrated that VX-445 specifically enhances the expression and maturation of the MSD2 domain. Furthermore, they showed that VX-445 exerts an additive effect on the functional expression of F508del CFTR when combined with either type 1 or type 2 CFTR correctors, effectively improving CFTR folding, trafficking, and stability [106].

#### 6.3.2. Other Type 3 Correctors

One of the first groups of CFTR correctors identified and developed in the 1990s was the benzo[c]quinolizinium (MPB) compound series [155]. Among these, MPB-07 demonstrated a remarkable ability to relocate the F508del CFTR protein to the PM, restoring its distribution in F508del/F508del CF cells to a pattern similar to that observed in cells expressing WT CFTR [156]. Further studies demonstrated that MPB-07 along with its derivative MPB-91, selectively prevents the proteolytic cleavage of F508del CFTR by directly binding to its NBD1 domain, thereby enhancing the stability and trafficking to the PM of F508del-CFTR in intact cells [157].

Odolczyk and co-workers employed structure-based virtual screening to identify low-molecular-weight compounds capable of binding to F508del-NBD1, specifically targeting the structural instability caused by the F508del mutation [158]. Among the identified compounds, the bisphosphonate compound C407 and its derivative G1 demonstrated significant potential in stabilizing NBD1 during the co-translational folding of CFTR. Specifically, C407 appears to function by occupying and mimicking the position of the Phe508 side chain, effectively compensating for the structural disruption caused by the deletion.

Despite promising results in cell lines, where C407, alone or in combination with VX-809, increased CFTR expression, these outcomes were not replicated in primary human respiratory cells carrying the F508del mutation. This limitation underscores the need for further optimization of these compounds to enhance their therapeutic potential and achieve more consistent and clinically relevant outcomes [159].

In the search for new CFTR correctors, Sampson and colleagues employed differential scanning fluorimetry to identify the phenylhydrazone compound RDR1, which exhibited moderate activity in rescuing F508del CFTR function in both cell-based models and an F508del CF mouse model [160]. Building on these findings, subsequent investigations of phenylhydrazone analogs, such as compounds 3c and 5d, revealed promising corrective activities at specific concentrations (20 µM and 2 µM, respectively) [161]. Notably, combining 5d with VX-809 produced additive effects, indicating the potential for synergistic benefits when phenylhydrazones are combined with clinically approved correctors.

Further research led to the identification of MCG1516A as a particularly promising phenylhydrazone compound. When used in combination with RDR1 and VX-809, MCG1516A restored F508del CFTR function to over 20% of that observed in non-CF control human bronchial epithelial cells and demonstrated even greater effectiveness in other cell types. MCG1516A was shown to act as a corrector for both WT and F508del-CFTR, exhibiting additive effects when combined with VX-661or the genetic revertant R1070W, suggesting its likely interaction with the NBD1:NBD2 interface. However, the lack of additivity with other revertants, such as G550E or the traffic-null variant p.Asp565Ala-Asp567Ala variant (DD/AA), implied a specific selectivity in its rescue mechanism. Additionally, MCG1516A in combination with VX-661 successfully rescued processing and functional activity for CFTR mutations like L206W and R334W, achieving significantly enhanced corrective outcomes [162,163].

Bamocaftor (VX-659) and vanzacaftor (VX-121) are next-generation correctors that share structural similarities with VX-445, justifying their classification as type 3 correctors. Bamocaftor, when combined with tezacaftor and ivacaftor, has been shown to significantly improve CFTR functional expression in CF patients carrying the F508del mutation. Importantly, the additive effects observed between bamocaftor and tezacaftor suggest that VX-659 binds to a different site from tezacaftor, highlighting their complementary mechanism of action [135]. In announcements from November 2018 and February 2024, Vertex Pharmaceuticals reported that Phase 3 clinical trials (NCT03911713 and NCT03912233), for two triple combination regimens—bamocaftor/tezacaftor/ivacaftor and vazacaftor/tezacaftor/deutivacaftor (“vanza triple”) regimen—showed statistically significant improvements in lung function (measured by ppFEV1) for CF patients. While these initial results appear promising, detailed findings from the trials are still awaited, as they are expected to provide critical insights into the mechanism of action, efficacy, and clinical potential of these advanced triple combination therapies.

In a 2023 study, Renda and collaborators introduced tricyclic pyrrolo-quinolines as novel correctors of the F508del-CFTR mutation, demonstrating high efficacy in primary airway epithelial cells derived from CF patients. Among these, PP028 emerged as the most potent candidate, showing synergistic effects with VX-809 or VX-661, but not with VX-445, and displaying antagonistic interactions with the type 3 corrector 4172. These findings suggest that PP028 operates through a mechanism of action similar to that of VX-445, highlighting its potential as a corrector suitable for combination therapies targeting CFTR processing defects [164]. Through iterative cycles of chemical synthesis and functional testing, the original library of pyrrolo-quinolines with corrector activity was further expanded leading to the discovery of a new class of molecules, the 6,9-dihydro-5H-pyrrolo[3,2-h]quinazolines, with promising potential for CFTR rescue. Within a series of 38 analogs, two derivatives, 3u and 3z, were identified as lead candidates and subjected to further analyses to elucidate their mechanism of action. Both compounds demonstrated a dose-dependent increase in CFTR functional expression, particularly when combined with VX-809. Their EC_50_ was in the single-digit micromolar range and decreased further in the presence of VX-809, suggesting a synergistic interaction with class 1 correctors. Synergy was also observed with corr4a, but not with VX-445 or PP028. This lack of synergy with other class 3 correctors strongly suggests that these tricyclic pyrrolo-quinazolines behave as class 3 correctors themselves [165].

In another 2023 study, Bacalhau and colleagues screened a collection of triazole compounds, identifying four promising hits, LSO-18, LSO-24, LSO-28, and LSO-39, which were validated for their ability to rescue F508del CFTR processing, promote plasma membrane trafficking, and restore channel function. Mechanistic evaluations revealed varying degrees of additivity with established correctors. Specifically, LSO-18, LSO-24, and LSO-28 were additive with VX-661, whereas LSO-28 and LSO-39 showed additive effects with VX-445. Furthermore, all compounds exhibited additivity with low-temperature treatment and genetic revertants such as G550E and 4RK but failed to rescue the DD/AA CFTR variant, suggesting that their mechanism of action is domain specific. The lack of additivity of LSO-39 with the R1070W genetic revertant suggests that this corrector targets a binding site at the NBD1:ICL4 interface, consistent with the binding characteristics of correctors like VX-661. In contrast, LSO-18 and LSO-24 appear to share a mechanism of action similar to VX-445, likely stabilizing inter-domain interactions such as those between NBD1 and the MSDs. Meanwhile, LSO-28 seems to operate through a distinct mechanism [166].

PTI-801 (posenacaftor) a third-generation CFTR corrector, has shown promise in enhancing F508del CFTR functional expression [167]. PTI-801 showed synergistic or additive interactions with other correctors, including ABBV-2222, FDL-169, VX-661, and VX-809, effectively enhancing defective CFTR processing, plasma membrane trafficking, and channel function. However, no additive effect was observed when PTI-801 was combined with VX-445, suggesting that these correctors may share a common binding site or allosteric pathway. The combination of PTI-801 with other correctors only partially restored the conformational stability of F508del CFTR, as evidenced by the shorter half-life of the mutant protein compared to WT CFTR, underscoring the need for further optimization [168].

A medicinal chemistry approach was utilized by Vaccarin and colleagues to design novel CFTR correctors based on the 4172 scaffolds originally identified by HTS by Veit and collaborators [133]. Through iterative optimization, four promising compounds—1, 10, 38, and 39—were identified. These derivatives demonstrated enhanced potency and efficacy in correcting the F508del CFTR misprocessing defect. Importantly, these optimized derivatives exhibited synergistic effects when combined with type 1 and type 1 correctors, significantly improving both plasma membrane density and functional expression of F508del CFTR. Moreover, these compounds also displayed efficacy in rescuing processing and trafficking defects in rare CFTR mutants that are resistant to currently approved therapies, suggesting their potential for inclusion in next-generation combinatorial treatments for CF [169].

**Table 5 cimb-47-00119-t005:** Investigational and approved type 3 correctors.

Name	Structure	Reference
4172N-[5-(1H-pyrazol-4-yl)-1H-pyrazol-3-yl]-2-{[2-(2-chlorophenyl)cyclopentyl]carbonyl}benzamide	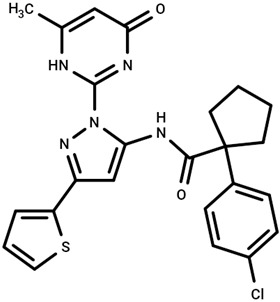	[133]
VX-445Elexacaftor*N*-(1,3-dimethylpyrazol-4-yl)sulfonyl-6-[3-(3,3,3-trifluoro-2,2-dimethylpropoxy)pyrazol-1-yl]-2-[(4*S*)-2,2,4-trimethylpyrrolidin-1-yl]pyridine-3-carboxamide	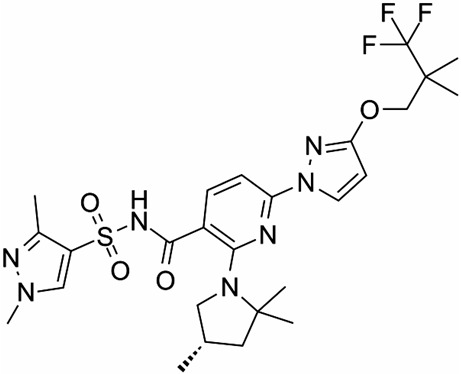	[58,102,106,134,135,136,137,138,139,140,141,142,143,144,145,146,147,148,149,150,151,152,153,154]
MPB-076-hydroxy-10-chlorobenzo(c)quinolizinium	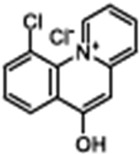	[155,156,157]
C407[hydroxy(phenoxy)phosphoryl]oxy(phenoxy)phosphinic acid	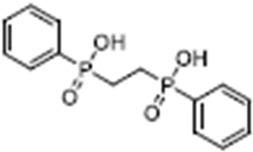	[158]
G1[4-(4-fluorophenoxy)phenyl]oxy-bis[(phenoxy)phosphoryl]methane	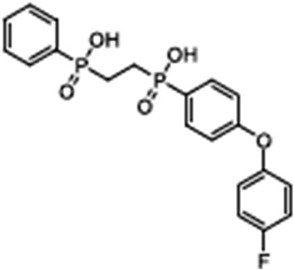	[159]
RDR1N-phenyl-1-(4-nitrophenoxy)-5-pyrazolamine	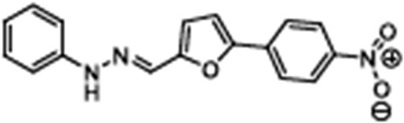	[160,161]
MCG1516A4-methyl-N-[3-(morpholin-4-yl) quinoxalin-2-yl] benzenesulfonamide	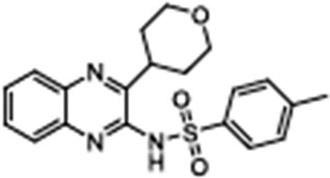	[162,163]
VX-659(Bamocaftor)N-(benzenesulfonyl)-6-[3-[2-[1-(trifluoromethyl)cyclopropyl]ethoxy]pyrazol-1-yl]-2-[(4*S*)-2,2,4-trimethylpyrrolidin-1-yl]pyridine-3-carboxamide	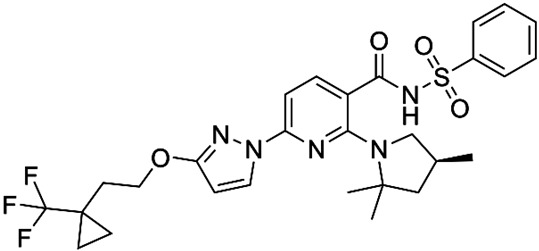	[135]
VX-121(Vanzacaftor)(14S)-8-[3-(2-dispiro[2.0.24.13]heptan-7-ylethoxy)pyrazol-1-yl]-12,12-dimethyl-2,2-dioxo-2lambda6-thia-3,9,11,18,23-pentazatetracyclo[17.3.1.111,14.05,10]tetracosa-1(22),5(10),6,8,19(23),20-hexaen-4-one	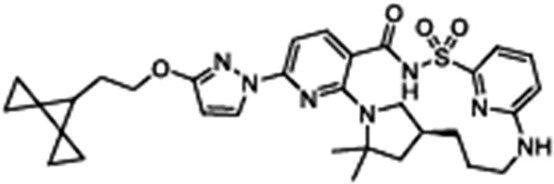	[135]
PP028Ethyl 7-bromo-6-(phenylsulfonyl)-1,2,3,4-tetrahydropyrrolo[4,3,2-de]quinoline-5-carboxylate	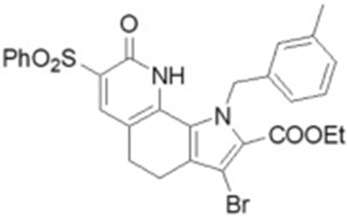	[164]
3c1-(4-Bromophenyl)-2-{[5-(4-nitrophenyl)furan-2-yl] methylene}hydrazine (3c)	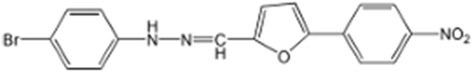	[165]
3d1-(2-Chlorophenyl)-2-{[5-(4-nitrophenyl)furan-2-yl] methylene}hydrazine (3d)	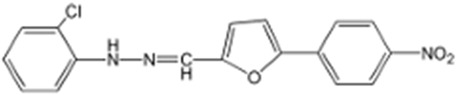	[165]
LSO-185-(4-bromophenyl)-3-(2-bromo-4-hydroxyphenyl)-1H-1,2,4-triazole derivative	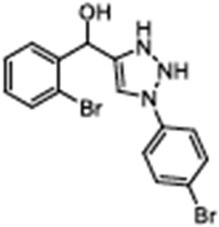	[166]
LSO-395-(4-bromophenyl)-3-(2-bromo-4-hydroxyphenyl)-1H-1,2,4-triazole derivative	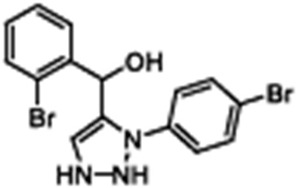	[166]
PTI-801 (Posenacaftor)8-Methyl-2-(3-methyl-1-benzofuran-2-yl)-5-[(1R)-1-(oxan-4-yl)ethoxy]quinoline-4-carboxylic acid	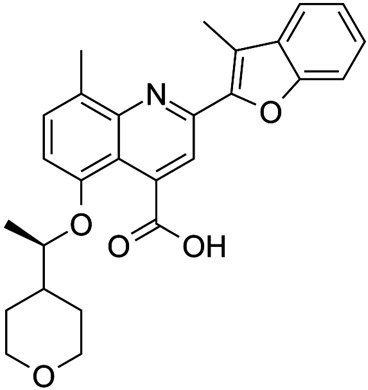	[167,168]

### 6.4. Type 1V Correctors

Recently, Marchesin and colleagues identified a novel class of CFTR correctors, referred to as type 4 correctors, that are characterized by a unique mechanism of action and binding site compared to previously characterized corrector types. By screening 67,772 compounds from the Idorsia chemical library, they developed four macrocyclic correctors (IDOR 1–4) that exhibited additive effects with known corrector types. These compounds promoted WT-like folding of the efficiency of F508del CFTR at the endoplasmic reticulum and restored native-like CFTR currents in patient-derived F508del bronchial epithelial cells. Employing advanced techniques, including photo-cross-linkable macrocycles, site-directed mutagenesis, and molecular modeling, the researchers identified a likely binding site for these correctors within a cavity between lasso helix-1 (Lh1) and the first two TM helices of MSD1. This binding site is distinct from those targeted by type 1, 2, and 3 correctors, aligning with the additive interactions observed when type 4 correctors were combined with existing modulators. Type 1V correctors appear to co-translationally stabilize interactions between Lh1, MSD1, and MSD2, significantly enhancing the folding efficiency of F508del CFTR. Importantly, these correctors demonstrated the ability to rescue other CFTR folding mutants, including I507del, R560T, V520F, and R1066C, which are unresponsive to tezacaftor (VX-661) and elexacaftor (VX-445) combination therapies, shrinking the spectrum of corrector-unresponsive CFTR mutations [170].

## 7. Conclusions and Future Directions

The development of CFTR modulators has revolutionized cystic fibrosis (CF) therapy, providing effective treatments for the majority of people with CF. Combination therapies, particularly triple regimens like elexacaftor/tezacaftor/ivacaftor (Trikafta/Kaftrio), have demonstrated significant improvements in lung function, chloride transport, and quality of life by addressing the protein misfolding and trafficking defects associated with the F508del CFTR mutation.

Building on this success, further research is being conducted on additional triple combinations currently under investigation in preclinical studies and clinical trials. As already mentioned, dual or triple combinations of modulators, combining a potentiator with one or two correctors, are being explored, including combinations such as GLPG-2451/GLPG-2222, deutivacaftor (VX-561, CTP-656)/vanzacaftor (VX-121)/tezacaftor (VX-661), dirocaftor (PTI-808)/posenacaftor (PTI-801)/nesolicaftor (PTI-428), and the the bamocaftor/tezacaftor/ivacaftor and vazacaftor/tezacaftor/deutivacaftor (“Vanza triple”).

These combinations aim to target different stages of CFTR biogenesis and function, ranging from protein folding to channel gating in an effort to provide a more comprehensive therapeutic effect. As these combinations progress through clinical trials, they hold great promise in broadening treatment options and offering new solutions for patients with rare and ultra-rare mutations or those who have not responded to existing therapies.

However, despite these remarkable advancements, challenges persist for patients with rare and ultra-rare CFTR mutations or premature termination codons that remain unresponsive to current therapies. Significant research efforts have focused on the identification and development of next-generation modulators, including type IV correctors and dual-acting compounds, which have shown promising efficacy in rescuing the functional expression of CFTR mutants with gating and processing defects. Additionally, mutation-agnostic approaches, such as gene editing and mRNA replacement therapies, are being explored to address mutations previously considered untreatable, potentially opening new treatment avenues for a broader patient population. Ongoing research into stabilizers, amplifiers, and proteostasis modulators—molecules not yet available on the market—continues to gain momentum. Moreover, activators, small molecules that regulate intracellular levels of cAMP or ATP to activate CFTR, are being deeply studied for their potential in CF treatment. Read-through agents, which target mutations causing premature termination codons (PTCs), are also being explored as a strategy to extend the functional expression of CFTR in patients with these mutations. Many of these modulators are under deep investigation to elucidate their mechanisms of action, and this research will hopefully provide crucial insights for the rational design of future drugs. Non-specific CFTR modulators, which do not directly bind to CFTR but exert their effects indirectly by targeting proteins involved in CFTR interactions or modulating CFTR-related mRNA or DNA, also represent an area of growing interest. However, since these modulators tend to have broad mechanisms of action, affecting CFTR as well as multiple other cellular effectors, potential side effects must be carefully considered.

In addition to their pharmacological benefits, CFTR modulators are also being investigated for their impact on microbial and functional lung issues in cystic fibrosis patients. CF lung disease is characterized by chronic infection and inflammation, with common pathogens such as *Pseudomonas aeruginosa* and *Burkholderia cepacian*, contributing to disease progression. By improving CFTR function, these therapies could help restore mucociliary clearance, reduce bacterial load, and alleviate airway inflammation. Moreover, modulator therapies may improve immune responses and the lung microenvironment, potentially helping manage chronic infections and improving lung function over time. While this area of study is outside the primary scope of this review, it was important to mention the broader therapeutic potential of CFTR modulators—not only as treatments targeting the basic CFTR defect but also as modulators of the lung environment, helping to address the multifaceted challenges of CF lung disease [171,172,173].

From the perspective of providing a rapid and effective response to clinicians’ demands, a key strategy to address current limitations in CF treatment is theratyping—a precision medicine approach that uses preclinical testing to predict the responsiveness of rare and ultra-rare CFTR mutations to existing or novel CFTR modulators. This strategy has already led to expanded approvals of specific modulators for certain mutations and continues to guide the development of personalized treatments.

The application of advanced techniques, such as cryo-electron microscopy (cryo-EM), molecular docking, and biochemical and functional assays, has provided key insights into modulator–CFTR interactions, enabling the development of next-generation correctors and potentiators with improved specificity, stability, and safety profiles. Deciphering the mechanisms of action and binding sites of CFTR modulators has proven critical for optimizing drug design and enhancing the therapeutic efficacy of new modulator scaffolds.

Looking ahead, all these advancements represent a concrete opportunity to expand treatment eligibility and address the unmet therapeutic needs of CF patients. With continued collaboration among academia, the industry, and patient organizations, the goal of personalized therapies, and, potentially, a cure for all CF patients, becomes an achievable reality.

## Figures and Tables

**Figure 1 cimb-47-00119-f001:**
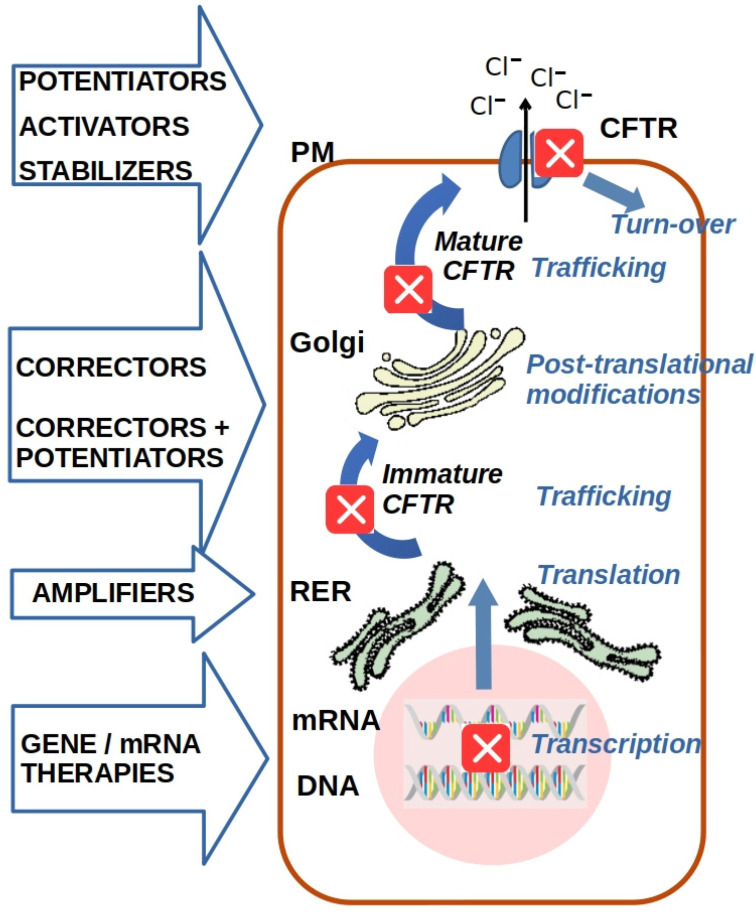
Schematic overview of the therapeutic approaches for causal treatment of CF. Investigational and currently available drugs for the treatment of CF have been developed to target the basic defect in CFTR. They act at various points in the CFTR biogenesis and maturation pathway. The beneficial effect of these molecules can be enhanced when administrated in combination, as the combination addresses different defects of the CFRT protein.

**Figure 2 cimb-47-00119-f002:**
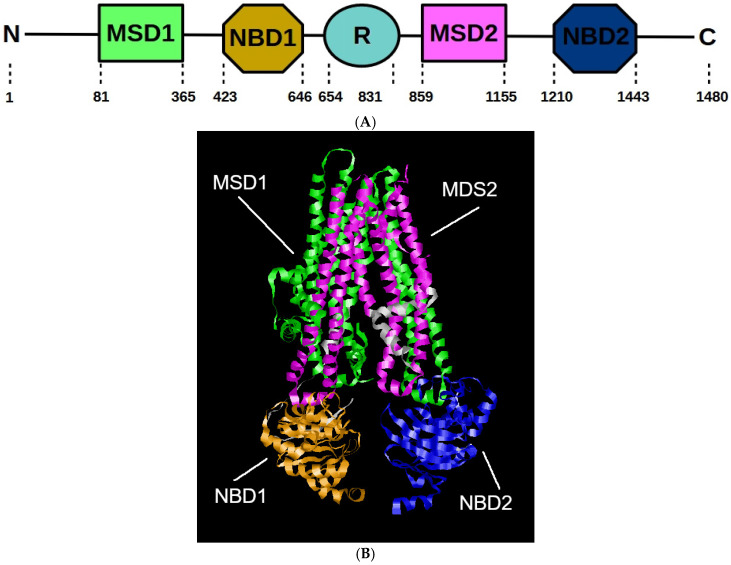
(**A**) Schematic representation of the CFTR domains. Different colors indicate the five domains of the protein: nucleotide-binding domain 1 (NBD1) in green, membrane-spanning domain 1 (MSD1) in yellow, regulatory domain (R domain) in cyan, membrane-spanning domain 2 (MSD2) in magenta, and nucleotide-binding domain 2 (NBD2) in blue. The numbers indicate the boundaries of each domain. (**B**) Structural model of the dephosphorylated, ATP-free human CFTR based on the coordinates from the 5UAK PDB entry, visualized using RASMOL (www.rasmol.org, accessed on 29 November 2024). The protein is depicted as colored ribbons following the same color scheme as in panel (**A**). The R domain is not included in this model.

**Figure 3 cimb-47-00119-f003:**
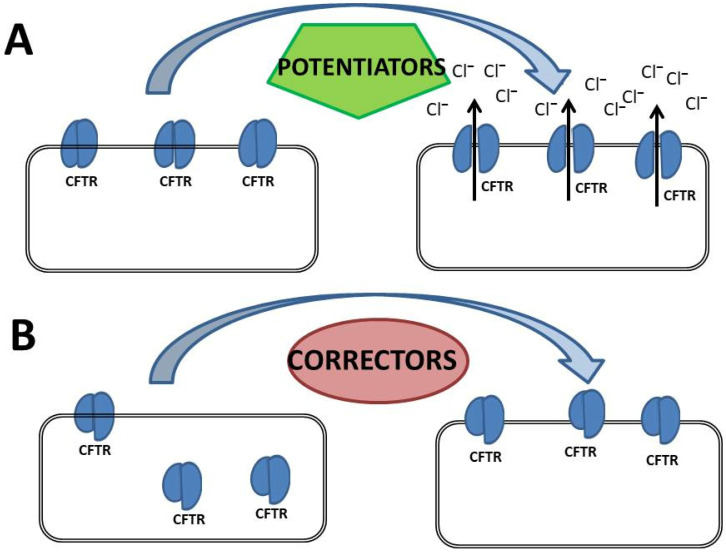
Impact of potentiators and correctors on CFTR functional expression: (**A**) Potentiators interact directly with the protein at the PM to enhance gating by increasing the channel’s open probability. (**B**) Correctors target CFTR promoting proper folding and improving trafficking to the plasma membrane.

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
