# Peer review of "Unraveling the Mechanism of Action, Binding Sites, and Therapeutic Advances of CFTR Modulators: A Narrative Review"

_cimb, 2025, doi:10.3390/cimb47020119_

Round 1
Reviewer 1 Report
Comments and Suggestions for Authors
The manuscript titled "Unraveling the Mechanism of Action, Binding Sites, and Therapeutic Advances of CFTR Modulators: A Narrative Review" provides a thorough and well-organized review of CFTR modulators, focusing on their mechanisms of action, binding sites, and advancements in cystic fibrosis treatment. It effectively highlights approved therapies such as Trikafta and explores ongoing research into investigational drugs, offering valuable insights. In my view, the manuscript is well structured and suitable for publication after minor revisions.
My suggestions for the authors are as follows:
- Please provide more details on how the studies included in the review were selected and analysed to improve clarity.
- If possible, include clear and illustrative figures or diagrams to better explain the mechanisms, binding sites, and interactions of CFTR modulators.
- Improve the quality of the 2D chemical structures to make them clearer and more visually appealing.
- Consider expanding the discussion to include additional investigational modulators that are in the early stages of development for a more balanced perspective.
- Clearly outline research gaps and propose future directions, such as strategies for addressing rare CFTR mutations.
Author Response
REVIEWER 1
Comment 1: The manuscript titled "Unraveling the Mechanism of Action, Binding Sites, and Therapeutic Advances of CFTR Modulators: A Narrative Review" provides a thorough and well-organized review of CFTR modulators, focusing on their mechanisms of action, binding sites, and advancements in cystic fibrosis treatment. It effectively highlights approved therapies such as Trikafta and explores ongoing research into investigational drugs, offering valuable insights. In my view, the manuscript is well structured and suitable for publication after minor revisions.
Response 1: I sincerely thank the reviewer for his thoughtful and constructive comments. I greatly appreciate the positive feedback regarding the quality of the manuscript and I am grateful for the time and effort dedicated to reviewing my work. Reviewer’s insightful suggestions have been invaluable in improving the clarity and comprehensiveness of the manuscript.
My suggestions for the authors are as follows:
Comment 2:: Please provide more details on how the studies included in the review were selected and analysed to improve clarity.
Response 2: Thank you for this valuable suggestion. Following the introduction section, I have added a detailed description of the selection and analysis process for the studies included in the review. Specifically, the criteria used to select relevant articles are now outlined, including the databases searched, the inclusion parameters, and the process of analysis and synthesis of the information. I believe this addition will enhance the clarity and transparency of the methodology.
Comment 3: If possible, include clear and illustrative figures or diagrams to better explain the mechanisms, binding sites, and interactions of CFTR modulators.
Response 3: Thank you for the suggestion. In response to your request, I have created a new figure (Figure 1) that clearly illustrates the mechanisms and interactions of CFTR modulators. The figure has been designed to visually represent how modulators intervene to rescue CFTR defects at various stages of its biogenesis and maturation pathway, helping to clarify their mode of action, including when used in combination.
Comment 4: Improve the quality of the 2D chemical structures to make them clearer and more visually appealing.
Response 4: Thank you for your helpful suggestion. In response, I have improved the quality of the 2D chemical structures to enhance their clarity and visual appeal. The updated images have been generated with higher resolution, with more precise detailing, to ensure they are clearer and more visually informative for readers.
Comment 5: Consider expanding the discussion to include additional investigational modulators that are in the early stages of development for a more balanced perspective.
Response 5: As suggested, I have expanded the conclusion section mentioning some of the most promising investigational modulators that are currently in the preclinical stages of development. I have highlighted their potential therapeutic targets and future prospects. I believe this addition provides a more complete and balanced picture of the current research landscape of CF modulators.
Comment 6:: Clearly outline research gaps and propose future directions, such as strategies for addressing rare CFTR mutations.
Response 6: In the revised manuscript, I added a paragraph in the final section that clearly outlines the major research gaps and future directions for CF therapeutic strategies. I discussed various types of molecules currently under investigation, including stabilizers, amplifiers, activators, read-through agents, as well as gene editing and RNA-based therapies. Additionally, I highlighted theratyping as a valuable strategy for testing combinatory modulator therapies. By focusing on these strategies, it may be possible to expand therapeutic eligibility and improve outcomes for patients with rare and ultra-rare CFTR mutations.
Reviewer 2 Report
Comments and Suggestions for Authors
The review is very interesting and complete but is too long and has to be shortenerd. The description of all the modulators is too detailed with all the molecules, even those that has proved not efficient, listed and their mechanism of presumptive action or binding to CFTR sites. I think it would better to focus on the most important, from the clinical point of view or representing most important noveltirs, modulators and to highlight on the combinations that gave better results and are used in the clinical treatment. The tables are giving the names of all the modulators and it seems not to be necessary to descrive each in detail.
The conclusions and future directions are well described focusing in the most interesting aspects of the review.
I suggest to treat something more the interaction among the molecules in combination as well as the studies regarding implementation on the microbial and functional lung problems.
Author Response
Reviewer 2
Open Review
Comments and Suggestions for Authors
Comment 1: The review is very interesting and complete but is too long and has to be shortenerd. The description of all the modulators is too detailed with all the molecules, even those that has proved not efficient, listed and their mechanism of presumptive action or binding to CFTR sites. I think it would better to focus on the most important, from the clinical point of view or representing most important novelties, modulators and to highlight on the combinations that gave better results and are used in the clinical treatment.
Response1: I would like to sincerely thank the reviewer for the thoughtful feedback. The detailed and lengthy description of the mechanism of action and binding sites for each drug were intentionally included to emphasize the complexity and the considerable effort involved in identifying and developing effective CFTR modulators. My aim by providing these insights was to highlight the challenges researchers have faced in understanding the molecular mechanisms and the long path leading to the development of successful therapies. I hope that this level of detail helps readers appreciate the intricacy of the process and the scientific rigor involved in developing CFTR modulators.
However, in accordance with the reviewer’s suggestions, I have shortened some descriptions of older drugs and those that have not shown efficacy, focusing more on the modulators that have demonstrated significant progress. I hope that this revision ensures that the manuscript remains concise while still conveying essential information.
Comment 2: The tables are giving the names of all the modulators and it seems not to be necessary to descrive each in detail.
Response2: Thank you for your comment. In the text, I intentionally avoided including the chemical names of most of the modulators mentioned, as they were deemed unnecessary for the general discussion and would have added unnecessary complexity to the narrative. I appreciate your suggestion to remove the chemical names from the tables, but I would prefer to retain them in the tables, where they can be easily referenced by readers who wish to access this specific information. This approach ensures that the main text remains focused on the key concepts, while still providing the necessary details in an accessible format within the tables.
I have revised the tables and deleted molecules that have shown limited efficacy or are considered outdated. Additionally, when there were similar compounds from a same class of compounds, I left in the table the most promising CFTR corrector from that class, ensuring thst the tables maintain the readability while providing comprehensive details in an accessible format.
Additionally, I have improved the quality of the images of the chemical structures of the modulators in the tables, as the previous version was not sufficiently legible.
I hope these revisions meet your expectations and contribute to a more streamlined and informative presentation.
Comment 3: The conclusions and future directions are well described focusing in the most interesting aspects of the review.
I suggest to treat something more the interaction among the molecules in combination as well as the studies regarding implementation on the microbial and functional lung problems.
Response 3: Thank you for your valuable feedback. I appreciate your positive remarks on the conclusions and future directions, and I agree that further elaboration on the interaction among molecules in combination, as well as their impact on microbial and functional lung issues, would enhance the manuscript. In response to your suggestions, I have added a figure (Figure 1) to better illustrate the interactions and the action points of different modulators at various stages of CFTR maturation, both individually and in combination. This addition will help visualize the mechanisms of action and interactions more clearly. In the text, I have emphasized even more the importance of using combinations of modulators, as these combinations are specifically designed to address various CFTR defects and improve therapeutic outcomes.
Regarding the link between CFTR modulators and microbial/functional lung issues, I must clarify that this is not an area of expertise for me. Additionally, I feel it would be outside the scope of this review to delve deeply into this topic. However, I have included a brief comment on the potential impact of CFTR modulators on lung function and microbial issues in the conclusions section, recognizing their importance in these areas while keeping the focus on the main subject of the review.